# OBERON3 and SUPPRESSOR OF MAX2 1-LIKE proteins form a regulatory module driving phloem development

Eva-Sophie Wallner[1,6], Nina Tonn[1], Dongbo Shi ®[1,2,3,4], Laura Luzzietti[1], Friederike Wanke[5], Pascal Hunziker ®[1], Yingqiang Xu[1], Ilona Jung[1], Vadir Lopéz-Salmerón[1,7], Michael Gebert ®[1], Christian Wenzl[1], Jan U. Lohmann[1], Klaus Harter[5] & Thomas Greb ®[1]✉

Spatial specificity of cell fate decisions is central for organismal development. The phloem tissue mediates long-distance transport of energy metabolites along plant bodies and is characterized by an exceptional degree of cellular specialization. How a phloem-specific developmental program is implemented is, however, unknown. Here we reveal that the ubiquitously expressed PHD-finger protein OBE3 forms a central module with the phloem-specific SMXL5 protein for establishing the phloem developmental program in *Arabidopsis thaliana*. By protein interaction studies and phloem-specific ATAC-seq analyses, we show that OBE3 and SMXL5 proteins form a complex in nuclei of phloem stem cells where they promote a phloem-specific chromatin profile. This profile allows expression of *OPS*, *BRX*, *BAM3*, and *CVP2* genes acting as mediators of phloem differentiation. Our findings demonstrate that OBE3/ SMXL5 protein complexes establish nuclear features essential for determining phloem cell fate and highlight how a combination of ubiquitous and local regulators generate specificity of developmental decisions in plants.

The growth and body shape of multicellular organisms largely depend on a functional long-distance transport of energy metabolites to fuel stem cell activity. In vascular plants, sugars are photosynthetically produced in source organs, such as leaves, and delivered via the phloem to sink organs where they are allocated to storage tissues or stem cell niches, such as the root apical meristem (RAM)[1,2]. The dividing stem cells of the RAM are located next to a mostly dormant organizer, known as quiescent center (QC)[3]. These stem cells divide and differentiate in a strictly controlled manner to give rise to two phloem poles which ensure a steady energy supply to the RAM during root growth[4,5]. One phloem pole comprises a protophloem and a metaphloem strand, each forming a sieve element (SE) and a companion cell (CC) lineage[6]. During differentiation, SEs degrade most of their organelles to build connected sieve tubes for intracellular allocation of sugars, hormones, proteins and RNAs[7]. This is why functional SEs are metabolically sustained by CCs via intercellular channels named plasmodesmata[8]. Underlining the importance of the phloem, defects in protophloem development impair root growth, possibly, as a consequence of RAM starvation[4,9].

Due to the remarkable transition of phloem stem cells to cells holding an extreme degree of specialization, gaining insights into phloem formation and identifying its molecular regulators is highly instructive for our general understanding of cell fate regulation and differentiation[10–12]. Moreover, due to the importance of the phloem for

[1]Centre for Organismal Studies (COS), Heidelberg University, 69120 Heidelberg, Germany. [2]Japan RIKEN Center for Sustainable Resource Science (CSRS), Yokohama 230-0045, Japan. [3]Institute for Biochemistry and Biology (IBB), University of Potsdam, Potsdam 14476, Germany. [4]Japan Science and Technology Agency (JST), Saitama, Kawaguchi, Japan. [5]Center for Plant Molecular Biology (ZMBP), University of Tübingen, 72076 Tübingen, Germany. [6]Present address: Gilbert Biological Sciences, Stanford University, Stanford, CA 94305-5020, USA. [7]Present address: BD Bioscience, 69126 Heidelberg, Germany. ✉e-mail: thomas.greb@cos.uni-heidelberg.de

plant growth and physiology, revealing mechanisms of phloem formation holds great promises for crop production and may increase our understanding of plant evolution and of the adaptation to environmental conditions[13]. Importantly, although several genes, including *ALTERED PHLOEM DEVLEOPMENT* (*APL*), *OCTOPUS* (*OPS*), *BREVIS RADIX* (*BRX*), *BARELY ANY MERISTEM3* (*BAM3*), and *COTYLEDON VASCULAR PATTERN2* (*CVP2*) have been identified to regulate different aspects of phloem formation[4,9,11,14–17], those genes seem to act downstream of phloem specification leaving the question open of how a phloem-specific developmental program is initiated.

Recently, we revealed a central role of the SUPPRESSOR OF MAX2 1-LIKE (SMXL) protein family members SMXL3, SMXL4 and SMXL5 in phloem formation[5,18]. SMXL proteins are well-conserved nuclear-localized developmental regulators and, in *Arabidopsis thaliana* (Arabidopsis), form a protein family of eight members sub-divided into different sub-clades based on phylogeny and function[19–23]. Among those, SMXL6, SMXL7, and SMXL8 are proteolytic targets of the strigolactone signaling pathway which bind directly to promoter regions of downstream target genes and, thereby, repress their transcription[21,22,24–26]. In comparison, SMXL3/4/5 proteins act independently from strigolactone signaling as central regulators of phloem formation[5,27,28]. Their redundant and dose-dependent functions become obvious in double and triple mutants, which are completely deprived of protophloem formation within the RAM resulting in root growth termination a few days after germination[5]. Despite their fundamental role in phloem formation, the mechanism of SMXL3/4/5 protein action remained obscure.

In contrast to SMXL proteins whose activity is spatially highly restricted, OBERONs (OBEs) are a family of four ubiquitously expressed, nuclear-localized proteins essential for tissue specification and meristem maintenance starting from the earliest stages of embryo development[29–32]. This role is reflected by mutants deficient for either of the two *OBE* sub-families which are embryo lethal[30–32]. In the shoot apical meristem (SAM), *OBE3* (also known as *TITANIA1* (*TTA1*)), interacts genetically with the homeobox transcription factor gene *WUSCHEL* (*WUS*) in stem cell regulation[29]. In addition, *OBE1* and *OBE2* are associated with vascular patterning in the embryo[32]. Interestingly, OBEs carry a highly conserved plant homeodomain (PHD)-finger domain known to bind di- and trimethylated histone H3 which allows recruitment of chromatin remodeling complexes and transcription factors[33]. Indeed, OBE proteins show chromatin binding and remodeling activities important for root initiation during embryogenesis[30,31]. Taken together, OBEs have versatile roles associated with cell fate regulation in plants[29–32] but, as for SMXL proteins, their specific roles in distinct tissues and their mode of action is unknown.

Here, we report that OBE3 and SMXL5 proteins physically interact forming a functional unit during protophloem formation in the RAM. We provide evidence that SMXL5 and OBE3 proteins are instrumental for the establishment of a phloem-specific chromatin configuration and for the expression of phloem-associated regulators. By characterizing the *SMXL3/4/5-OBE3* interaction and of phloem-specific chromatin conformation, we provide insights into molecular mechanisms of cell specification and the establishment of a highly specialized and central plant tissue.

## Results

### *SMXL4* and *SMXL5* promote the expression of early phloem markers

To map the function of the *SMXL4* and *SMXL5* genes within the process of phloem formation, we introgressed a series of developmental markers visualizing early steps of phloem formation[34] into the *smxl4;smxl5* double mutants showing severe defects in protophloem formation[5]. Analysis of root tips 2 days after germination when the overall anatomy of the *smxl4;smxl5* RAM is comparable to wild-type

RAMs[5], showed that *OPS:OPS-GFP*, *BRX:BRX-CITRINE*, *BAM3:BAM3-CITRINE*, or *CVP2:NLS-VENUS* marker activities[4] were reduced or not detectable in *smxl4;smxl5* plants (Fig. 1a–h). This reduction was found along the entire strand of the developing protophloem and included SE-procambium stem cells located immediately proximal to the quiescent center (QC). In these founder cells of the phloem lineage, we observed accumulation of OPS-GFP and BRX-CITRINE fusion proteins in wild-type which was hardly detectable in *smxl4;smxl5* double mutants (Fig. 1i–p). Similar to markers associated with early stages of phloem development, the activity of the *APL* promoter marking differentiating SEs and CCs in wild-type[17] was not detectable in root tips of *smxl4;smxl5* mutants (Supplementary Fig. 1). These observations argued for an early stem cell-associated role of *SMXL4* and *SMXL5* in establishing a general phloem-specific developmental program including *OPS*, *BRX*, *BAM3*, and *CVP2* gene activities.

In contrast to the positive phloem regulators *OPS*, *BRX*, and *CVP2*, the phloem-associated BAM3/CLAVATA3/ESR-RELATED 45 (CLE45) receptor-ligand module counteracts phloem development[9,35]. We therefore tested whether a hyperactive BAM3/CLE45 signaling pathway is the reason for defective phloem development in *smxl4;smxl5* mutants. Arguing against this possibility, *smxl4;smxl5;bam3* triple mutants showed root growth defects similar to *smxl4;smxl5* double mutants and stimulating the BAM3 pathway by CLE45 treatments had no effect on *smxl4;smxl5* roots (Supplementary Fig. 1e). Together with the reduced BAM3 reporter activity (see above), this indicated that, like other phloem-related features, the phloem-associated BAM3/CLE45 pathway was less active in *smxl4;smxl5* mutants and not causing the observed developmental defects.

### Activity of *SMXL* genes upstream of *OPS* and *BRX* is required for phloem formation

To challenge the idea of an early role of *SMXL5*, we tested the capacity of *SMXL5* to restore protophloem formation when expressed under the control of promoters active during different phases of protophloem development[34]. Root length served as a fast and efficient read-out for phloem defects[5,9]. Supporting the need for *SMXL5* activity during early phases of phloem development, reduced root length and impaired SE formation usually found in *smxl4;smxl5* mutants were not observed when they expressed *SMXL5* under the control of the early *OPS*, *BAM3*, or *CVP2* promoters (Fig. 2a and Supplementary Fig. 1g–i). In contrast, driving *SMXL5* expression by the late *APL* promoter did not restore root length or SE formation (Fig. 2a and Supplementary Fig. 1j).

To see whether the reduced activity of regulators like *BRX* is causative for reduced root length of *smxl4;smxl5* double mutants, we expressed *BRX-VENUS* in *smxl4;smxl5* mutant backgrounds under the control of the protophloem-specific *SMXL4* promoter (Supplementary Fig. 1k)[5]. Indeed, the root length of *SMXL4:BRX-VENUS/smxl4;smxl5* lines was comparable to wild-type (Fig. 2a), indicating that *BRX* acts downstream of *SMXL4* and that reduced *BRX* activity is one reason for disturbed phloem development in *smxl4;smxl5* mutants. Visualization of SMXL4 and SMXL5 proteins in *OPS*-deficient backgrounds by respective fluorescent fusion proteins[5] did neither reveal reduced signal intensity nor altered localization of SMXL4 or SMXL5 proteins in protophloem cells (Fig. 2b–e). This suggested that in contrast to a positive effect of *SMXL4* and *SMXL5* on the activity of *OPS* and *BRX* genes (see above), *OPS* was not important for stimulating *SMXL4* or *SMXL5* activity and that *OPS* and *SMXL* genes function at distinct steps during phloem formation.

### *SMXL5* and *OPS/BRX* genes act on different steps of phloem formation

Our interpretation that *SMXL* genes act upstream of *OPS* and *BRX*, was confirmed by investigating their genetic interaction. The OPS protein is required for SE formation in the protophloem by counteracting the BAM3/CLE45 pathway[35]. Due to enhanced activity of the BAM3/CLE45

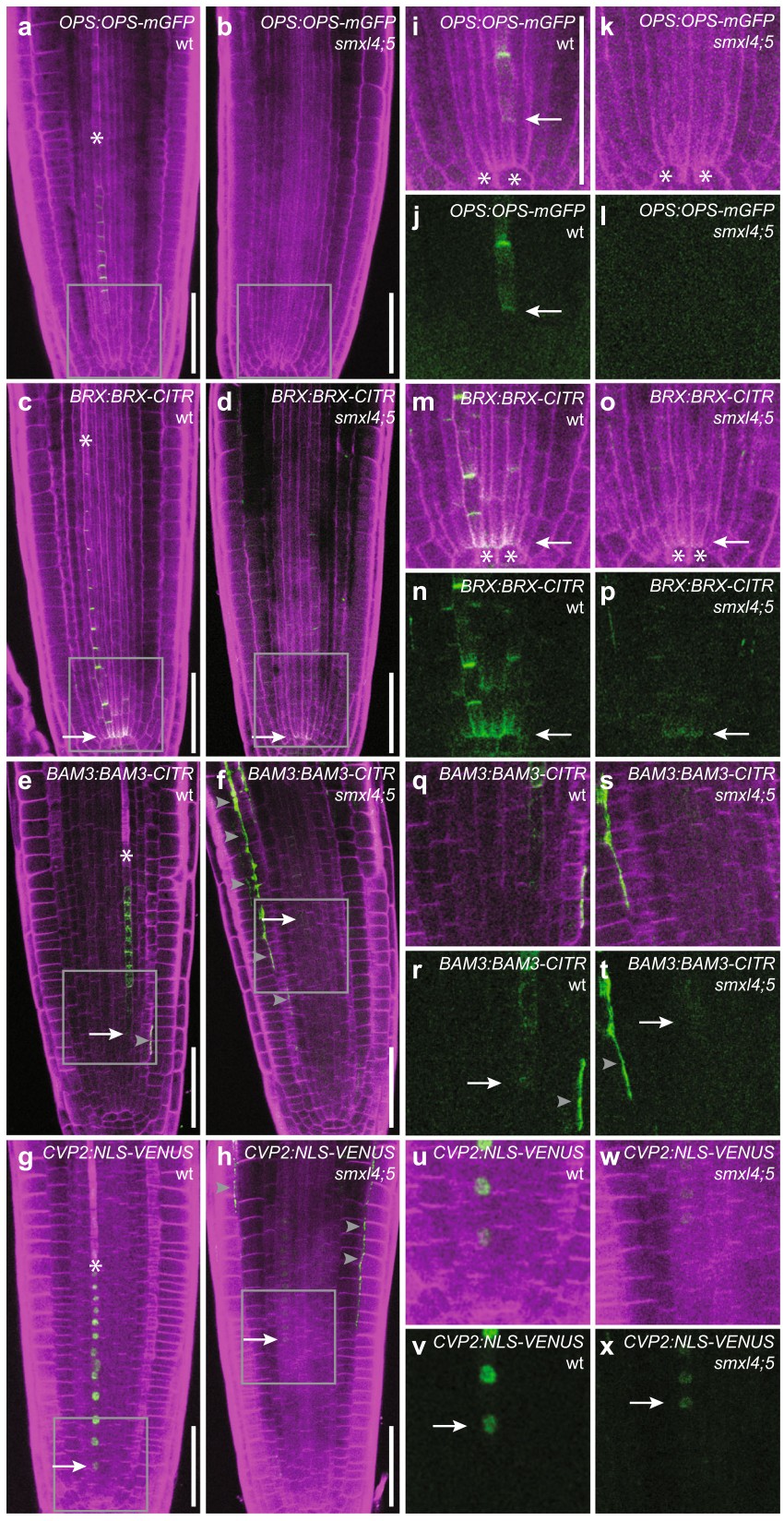

pathway, *ops* mutants develop 'gap cells' within protophloem strands in which SE formation fails[4,36]. Interestingly, *ops;smxl5* and *smxl4;smxl5* double mutants showed a similar reduction in root length although *smxl5* single mutants were not affected, and *ops* single mutants were very variable in this regard (Fig. 3a). This finding suggested an additive effect of *SMXL* and *OPS*-dependent pathways on phloem formation.

Indeed, when phloem development was carefully analyzed in the respective mutant backgrounds, we observed a variation of phloem defects in *ops* single mutants ranging from the appearance of gap cells to the complete absence of SEs in a small fraction of plants (Fig. 3b and Supplementary Fig. 2). In comparison, 60% of the *ops;smxl5* mutants displayed complete SE deficiency demonstrating that both genes

**Fig. 1 | Phloem-related transcriptional and translational reporters are less active in *smxl4;smxl5* mutants. a–h** Comparison of *OPS:OPS-GFP* (**a**, **b**), *BRX:BRX-CITRINE* (**c**, **d**), *BAM3:BAM3-CITRINE*, and *CVP2:NLS-VENUS* (**g**, **h**) reporter activities in wild-type (**a**, **c**, **e**, **g**) and *smxl4;smxl5* double mutants (**b**, **d**, **f**, **h**). Asterisks depict the first differentiating SE. Arrows point to earliest detectable reporter activities. Gray arrowheads indicate background signal due to tissue damage. Scale bars represent 50 μm. Ten samples each were analyzed with similar results.

**i–x** Magnification of the area indicated by gray rectangles in (**a–h**). *OPS:OPS-GFP* (**i–l**), *BRX:BRX-CITRINE* (**m–p**), *BAM3:BAM3-CITRINE* (**q–t**), and *CVP2:NLS-VENUS* (**u–x**) in wild-type (**i**, **j**, **m**, **n**, **q**, **r**, **u**, **v**) and *smxl4;smxl5* mutants (**k**, **l**, **o**, **p**, **s**, **t**, **w**, **x**). **j**, **l**, **n**, **p**, **r**, **t**, **v**, **x** Fluorescent signals are depicted without counterstaining. Asterisks indicate QC cells. Arrows point to the earliest detectable reporter activities. Gray arrowheads indicate background signal due to tissue damage. Scale bar in (**i**) represents 50 μm. Same magnification in (**i–x**).

contribute to robust phloem development. A similar trend was observed for *brx;smxl5* double mutants. Like *OPS*, the *BRX* gene ensures continuous SE formation, in this case, by downregulation of *BAM3* transcription and steepening the auxin gradient in developing phloem cells[9–11,37]. Similar to *ops;smxl5* double mutants, *brx;smxl5* plants developed shorter roots than *brx* and *smxl5* single mutants and largely failed to differentiate SEs (Fig. 3a and Supplementary Fig. 3). Importantly, in those *ops;smxl5* plants developing SEs, gap cell formation was comparable to *ops* and *brx* single mutants (Fig. 3b and Supplementary Fig. 1). Taken together, these observations suggested that *SMXL5* and *OPS/BRX* genes play roles at different steps during phloem formation with *SMXL5* acting upstream.

### SMXL5 proteins interact with OBE3 proteins in nuclei of plant cells

To understand how SMXL proteins fulfill their early role in phloem formation, we isolated interacting proteins by a Yeast-Two-Hybrid-based screen of a cDNA expression library generated from Arabidopsis seedlings[38,39] using the full-length SMXL5 protein as a bait. After testing 84 million individual protein–protein interactions, we identified OBE3 as a "very high confident" candidate interactor with 24 isolated independent cDNA clones (Supplementary Fig. 4). After confirming the yeast-based interaction of OBE3 with SMXL5 in independent experiments (Fig. 4a), we tested whether both proteins also interacted in planta. We transiently expressed SMXL5 fused to a triple human influenza hemagglutinin (HA) affinity tag and OBE3 fused to a sixfold c-Myc epitope tag in *Nicotiana benthamiana* (*N. benthamiana*) leaves under the control of the Cauliflower Mosaic Virus (CaMV) *35S* promoter[40]. In raw protein extracts before ("input") and after ('unbound') immunoprecipitation (IP) using HA-affinity beads and in the precipitate itself ("IP: α HA"), the SMXL5-3xHA protein was detected with the expected size of approximately 120 kDa in western blot analyses (Fig. 4b). Importantly, the 6xMyc-OBE3 fusion protein co-immunoprecipitated with the SMXL5-3xHA protein and did not show unspecific binding to the HA-affinity beads, indicating that SMXL5-3xHA and 6xMyc-OBE3 proteins interacted in plant cells.

To compare the subcellular localizations of SMXL5 and OBE3 proteins, we transiently expressed the SMXL5 protein fused to monomeric Cherry (SMXL5-mCherry) together with the OBE3 protein fused to monomeric GFP (OBE3-mGFP) again in *N. benthamiana* leaves. Initially, nuclear localization of the OBE3 protein was confirmed by co-expressing OBE3-GFP with mCherry fused to a nuclear localization signal (mCherry-NLS). Interestingly, while the mCherry-NLS signal was homogenously distributed within the nucleus, the OBE3-mGFP protein appeared in nuclear subdomains (Fig. 4c–e). Co-expression of SMXL5-mCherry and OBE3-GFP revealed a co-localization of both proteins within these domains (Fig. 4f–h) which were distinct from the whole nucleus highlighted by an mGFP-mCherry-NLS fusion protein expressed under the control of the *ubiquitin 10* (*UBQ10*) promoter (Fig. 4i–k). Next, we evaluated our yeast-two-hybrid and co-immunoprecipitation data by performing Förster resonance energy transfer (FRET)-fluorescence lifetime imaging microscopy (FLIM) analysis as an in planta assay for protein–protein association. In transiently transformed *N. benthamiana* epidermal leaf cells, FRET-FLIM analysis detected a significant decrease in the lifetime of the donor OBE3-mGFP fusions in the nucleus when co-expressed with SMXL5-mCherry (Fig. 4l, m). In contrast, we did not observe significant mGFP lifetime changes when

OBE3-mGFP was co-expressed with NLS-mCherry (Fig. 4l, m). Taking these observations together, we concluded that OBE3 interacts directly with SMXL5 in plant cell nuclei.

### The *OBE3* gene and the *SMXL3*, *SMXL4*, and *SMXL5* genes act together

Since physical interaction and subcellular co-localization suggested a common action of SMXL5 and OBE3 proteins, we investigated whether the corresponding genes are functionally connected by again using root length as a first read-out for potential phloem defects. As before, *smxl4;smxl5* double mutants were short-rooted, while root lengths of *smxl4* and *smxl5* single mutants were similar to wild-type[5] (Fig. 5a, b). Likewise, roots from *obe1*, *obe2*, *obe3*, and *obe4* single mutants resembled wild-type roots. In contrast, *smxl4;obe3*, *smxl5;obe3,* and *smxl3;obe3* double mutants had short roots just as *smxl4;smxl5* (Fig. 5a–d), suggesting a concerted action of *OBE3* and *SMXL3*, *SMXL4*, or *SMXL5* genes during primary root growth. In addition, roots of *smxl4;smxl5;obe3* triple mutant seedlings showed the same growth reduction as *smxl4;smxl5* and *smxl5;obe3* double mutants (Fig. 5e, f), demonstrating that, at this stage, growth defects could not be further increased by eliminating another member of this regulatory group. Of note, we only detected short roots when combining *SMXL3/4/5* and *OBE3* deficiency and not when combining *SMXL4/5* deficiency and deficiency in other *OBE* family members (Supplementary Fig. 5). In addition to 24 *OBE3* clones, nine clones of *OBE2* and three clones of *OBE4* in our initial yeast-two-hybrid screen (Supplementary Data 1) and SMXL5 and OBE2 proteins interacted in yeast cells in independent experiments (Supplementary Fig. 5). This indicated a general potential of SMXL5 to interact with OBE proteins, but a functional specificity of *OBE3* in phloem formation.

### *OBE3* locally promotes early phloem development

Because the reduced root length of *smxl;obe3* double mutants suggested a role of *OBE3* in phloem development, we next tested whether *OBE3* is expressed in developing phloem cells by comparing the activity pattern of a translational *SMXL4:SMXL4-YFP* reporter[5] with patterns of a translational *OBE3:OBE3-GFP* reporter[31] (Fig. 6a–d). As reported previously[5], the SMXL4-YFP protein accumulated specifically in nuclei of the protophloem lineage identified by enhanced Direct Red 23 staining (Fig. 6a). In comparison, the *OBE3:OBE3-GFP* reporter revealed OBE3-GFP protein accumulation in nuclei of all cell types of the root tip including developing protophloem cells (Fig. 6b) expressing SMXL3, SMXL4, and SMXL5 proteins (Fig. 6a)[5]. We thus concluded that SMXL3/4/5 and OBE proteins had the potential to interact during early phases of phloem formation. We could not detect differences in activity patterns between *OBE3:OBE3-GFP* and *OBE4:OBE4-GFP*[31] reporters (Fig. 6b, c) which argued against the possibility that differences in expression are the reason why *OBE3*, but not *OBE4*, genetically interacted with *SMXL3/4/5* genes.

To evaluate whether growth defects observed in *smxl;obe3* roots are correlated with the same type of protophloem defects observed in *smxl4;smxl5* mutants, we analyzed phloem development in the respective genetic backgrounds. During protophloem development, SE procambium-precursors divide periclinally to give rise to procambium and SE precursor cells. After 2–3 anticlinal divisions, SE precursor cells divide again periclinally to initiate meta- and protophloem cell files that subsequently undergo gradual differentiation

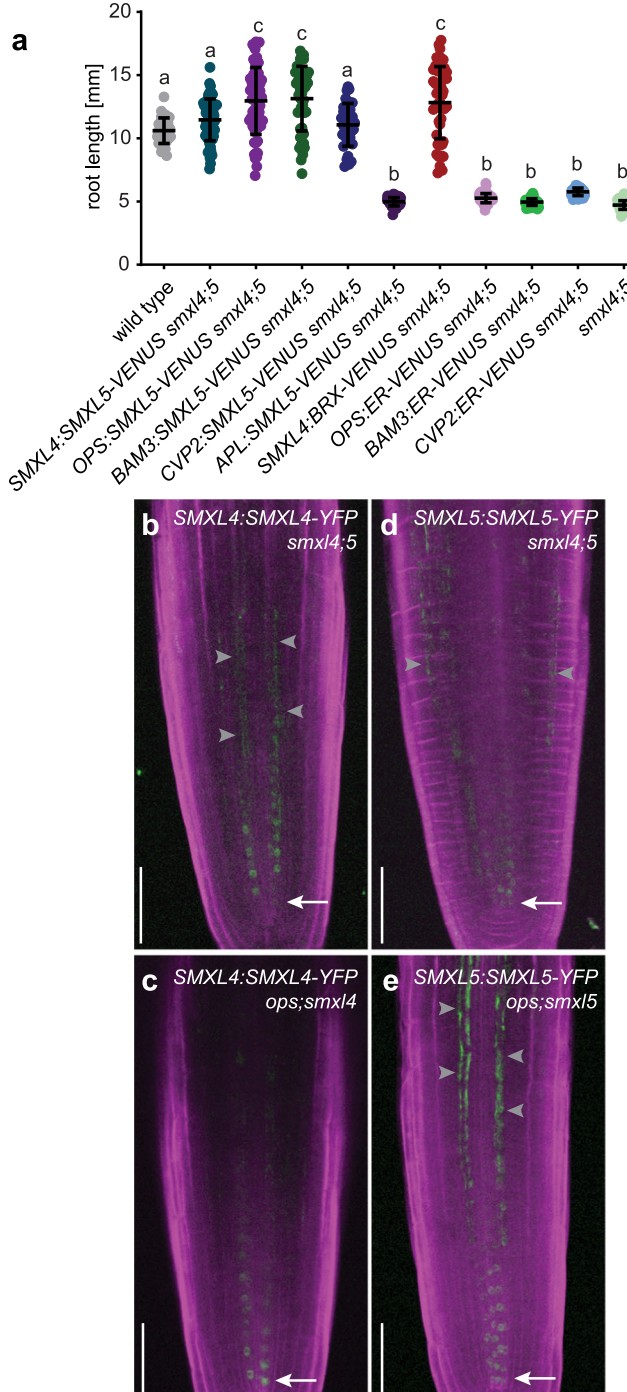

**Fig. 2 | Analysis of interaction between *SMXL4*, *SMXL5*, and other phloem regulators. a** Root length of 5-day-old plants. *n* between 36 (for *CVP2:SMXL5-VENUS;smxl4;5*) and 72 (for *APL:SMXL5-VENUS;smxl4;5*), see Source Data File for exact sample sizes of all groups. Statistical groups determined by one-way ANOVA and post hoc Tukey's test (95% CI). Shown is one representative experiment of three repetitions. **b–e** Comparison of *SMXL4:SMXL4-YFP* and *SMXL5:SMXL5-YFP* reporter activities in *smxl4;smxl5* and *smxl;ops* mutants. Arrows indicate the signals closest to the QC. Gray arrowheads indicate background signal due to tissue damage. Scale bars represent 50 μm. Ten samples each were analyzed with similar results.

(Fig. 6d)[5]. Our analysis revealed that in *obe3* mutants both the periclinal cell divisions and the onset of SE differentiation appeared as in wild-type roots (Fig. 6e, f, k, l). In contrast, in *smxl4;obe3* and *smxl5;obe3* mutants, the onset of the second periclinal division initiating meta- and protophloem cell lineages was similarly delayed as in *smxl4;smxl5*

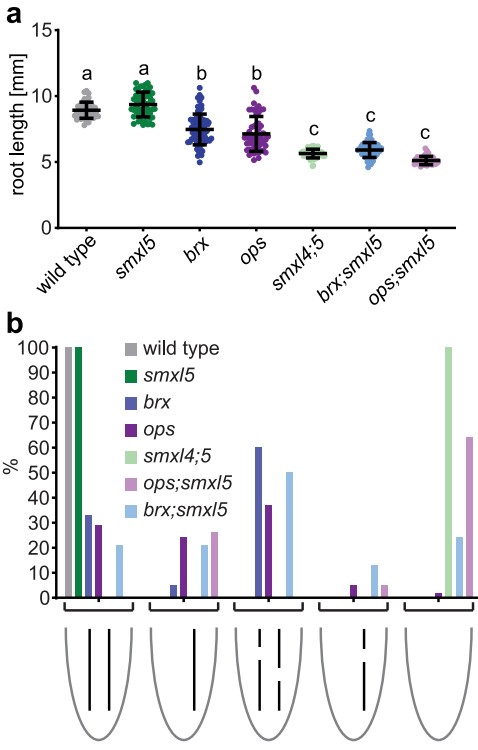

**Fig. 3 | Genetic interaction of *SMXL5* with *OPS* and *BRX*. a** Root length of 5-day-old wild-type (WT), *smxl5*, *brx*, *ops*, *smxl4;smxl5*, *brx;smxl5*, and *ops;smxl5* plants. *n* between 38 (for *ops;smxl5*) and 76 (for *brx*), see Source Data File for exact sample sizes of all groups. Statistical groups determined by one-way ANOVA and post hoc Tukey's HSD test (95% CI). Shown is one representative experiment of three repetitions. **b** Phenotypic characterization of phloem development of 2-day-old wild-type, *smxl5*, *brx*, *ops*, *smxl4;smxl5*, *brx;smxl5*, and *ops;smxl5* plants. *n* between 18 (for s*mxl5* and *brx*) and 49 (for *brx;smxl5*), see Source Data File for exact sample sizes of all groups.

mutants and the enhanced mPS-PI staining visualizing differentiated SEs was likewise absent (Fig. 6g–i, k, l and Supplementary Fig. 6). This observation demonstrated that, like the more locally expressed *SMXL3/4/5* genes, *OBE3* substantially contributes to phloem formation. Moreover, because respective single mutants did not show these defects (see above and Supplementary Fig. 6), we concluded that *OBE3* or *SMXL5*-deficient plants represent sensitized backgrounds for the functional loss of the other regulator. In accordance with the wild-type-like root growth of those plants, protophloem formation in *smxl5;obe4* double mutants was indistinguishable from wild-type (Fig. 6e, j, l).

Because *OBE3* is broadly expressed, phloem defects observed in *smxl5;obe3* mutants could arise due to a function of *OBE3* in other tissues than developing phloem cells, which would contradict a direct interaction of SMXL5 and OBE3 proteins. To address this concern and to determine whether *OBE3* acts cell-autonomously on phloem development, we expressed *OBE3* exclusively in developing phloem cells by introducing a transgene driving an OBE3-turquoise fusion protein under the control of the *SMXL5* promoter (*SMXL5:OBE3-turquoise*) into a *smxl5;obe3* double mutant background. Microscopic analysis of root tips from *smxl5;obe3/SMXL5:OBE3-turquoise* lines confirmed the presence of the OBE3-turquoise protein in nuclei of developing protophloem cells (Fig. 7a, b) as described for the SMXL5 protein expressed under the control of the same promoter[5]. When comparing *smxl5;obe3/SMXL5:OBE3-turquoise* lines with *smxl5;obe3* mutants, we observed that expression of *OBE3-turquoise* within the *SMXL5* domain was indeed sufficient to restore root length in *smxl5;obe3* double mutants (Fig. 7c, d). In addition, enhanced Direct Red staining of the mature protophloem indicating SE differentiation

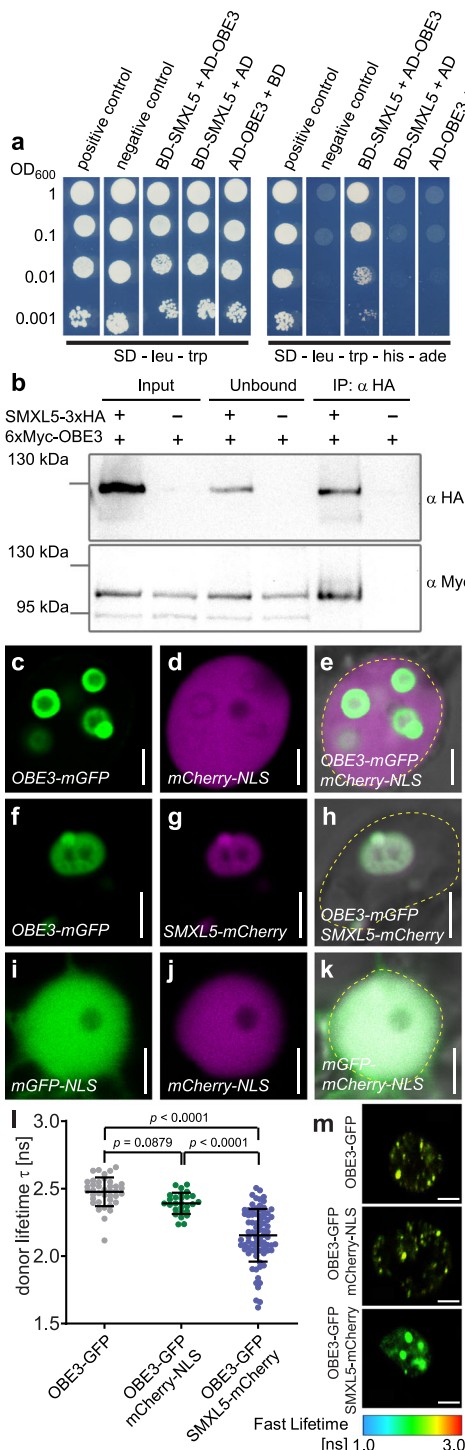

**Fig. 4 | SMXL5 and OBE3 proteins interact. a** The SMXL5 protein was expressed in yeast fused to the GAL4 DNA-binding domain (BD) and OBE3 fused to the GAL4 activation domain (AD). All strains contained AD- and BD-expressing plasmids either alone or fused with SMXL5, OBE3 or control proteins, respectively. Growth on SD-leu-trp indicate the presence of both plasmids, growth on SD-leu-trp-his-ade medium indicate the presence of plasmids and protein interaction. $OD_{600}$ values indicate the density of spotted yeast cultures. Positive control: yeast strain expressing SV40 large-T-antigen fused to GAL4-AD and p53 fused to GAL4-BD, negative control: yeast strain expressing lamin C fused to GAL4-BD and the GAL4-AD. **b** Interaction of SMXL5-3xHA and 6xMyc-OBE3 proteins by co-immunoprecipitation (co-IP) and subsequent western blot analysis after transient overexpression in *N. benthamiana*. "Input" represents unprocessed protein extracts, "unbound" shows proteins that remained in the extract after IP and "IP: α HA" depicts samples after immunoprecipitation by α-HA-beads. Western blots were probed by α HA or α Myc antibodies, respectively. Signals revealed an expected SMXL5-3xHA protein size of ~120 kDa. The size of the detected 6xMyc-OBE3 protein (~100 kDa) exceeded the expected size of 92 kDa. This experiment was repeated twice. Uncropped blots are included in the Source Data File. **c–k** Fluorescent signals and bright-field images of epidermal *N. benthamiana* nuclei transiently co-expressing OBE3-mGFP/mCherry (**c–e**), OBE3-mGFP/SMXL5-mCherry (**f–h**) and mGFP-NLS/mCherry-NLS (**i–k**). The dashed yellow line indicates the outlines of nuclei in merged images (**e, h, k**). Scale bars represent 5 μm. **l** FRET-FLIM analysis of transiently transformed *N. benthamiana* epidermal leaf cells expressing the OBE3-mGFP donor without an mCherry acceptor or in the presence of NLS-mCherry or SMXL5-mCherry. Error bars indicate standard deviation. Data were derived from three biological replicates with *n* = 27–80. To test for homogeneity of variance, a Brown–Forsythe test was applied. As the variances are not homogenous a Wilcoxon/Kruskal–Wallis test was carried out followed by a Dunn post hoc analysis. **m** Heatmaps of representative nuclei used for FLIM measurements. The donor lifetimes of OBE3-mGFP are color-coded according to the scale at the bottom. Size bars represent 4 μm.

(Supplementary Fig. 6), demonstrating that the activity of *OBE3* and *SMXL5* is necessary but not sufficient for executing phloem development.

## *SMXL* and *OBE3* genes determine chromatin structure in phloem cells

To probe the common role of SMXL and OBE3 proteins and a putative effect of their absence on chromatin signatures[25,26,31], we performed phloem-specific Assays for Transposase-Accessible Chromatin using sequencing (ATAC-seq) revealing chromatin structure in a genome-wide fashion[42]. To this end, we fluorescently labeled nuclei in phloem-associated cells in wild-type, *smxl5*, *smxl4;smxl5* and *smxl5;obe3* plants by expressing a histone H4-GFP protein fusion under the control of the *SMXL5* promoter (*SMXL5:H4-GFP*[43], Supplementary Fig. 7a). Isolation of nuclei from those lines and their fluorescence-activated sorting into GFP-positive and GFP-negative populations (FANS[44], Supplementary Fig. 7b) allowed ATAC-seq analyses on phloem-related and non-phloem-related cells separately. Thereby, we detected 36,566 to 40,168 open chromatin regions (OCRs) in all sample types mostly located in 5' and 3' regions of respective gene bodies (Fig. 8a, b and Supplementary Figs. 8–10). Importantly, the overall conformation was similar in all samples (Supplementary Fig. 8) suggesting, by large, a comparable chromatin structure in phloem and non-phloem cells and an independency of most chromatin domains from *SMXL4*, *SMXL5* or *OBE3* activity. Among 38,562 OCRs detected in phloem cells from wild-type, 1802 OCRs associated with 1311 genes showed a significant increase of read alignment in comparison to non-phloem cells (Fig. 8a and Supplementary Data 2 and 3; fold change >2, poisson enrichment *P* value <0.05, using 40 M reads). These genes included OCRs in promoter regions of the phloem-specific *OPS*, *BAM3*, *CVP2*, and *APL* genes (Supplementary Figs. 9 and 10 and Supplementary Data 2 and 3), suggesting that we succeeded in phloem-specific ATAC-seq analysis. In accordance with the wild-type-like phenotype of *smxl5* mutants, 1511 of 36,925 OCRs associated with 1139 genes showed a significant increase

was recovered in *smxl5;obe3* carrying the *SMXL5:OBE3-turquoise* transgene (Fig. 7a, b). To see whether the predominant reduction of *OBE3* activity in developing phloem cells is furthermore sufficient for generating phloem defects, we designed two artificial microRNAs targeting the *OBE3* mRNA (*obe3-miRNAs*)[41] and expressed them independently under the control of the *SMXL5* promoter in *smxl5* mutant plants. As expected, the majority of those plants was short-rooted, indicating that *OBE3* knock-down within the *SMXL5* domain was sufficient to evoke a *smxl5;obe3*-like phenotype in *smxl5* mutants (Fig. 7c, d). We thus concluded that *OBE3* fulfills a cell-autonomous and *SMXL5*-dependent role in protophloem formation. Interestingly, ectopic expression of *SMXL5* using the *35S* promoter did, although *OBE3* is broadly expressed, not lead to ectopic phloem formation

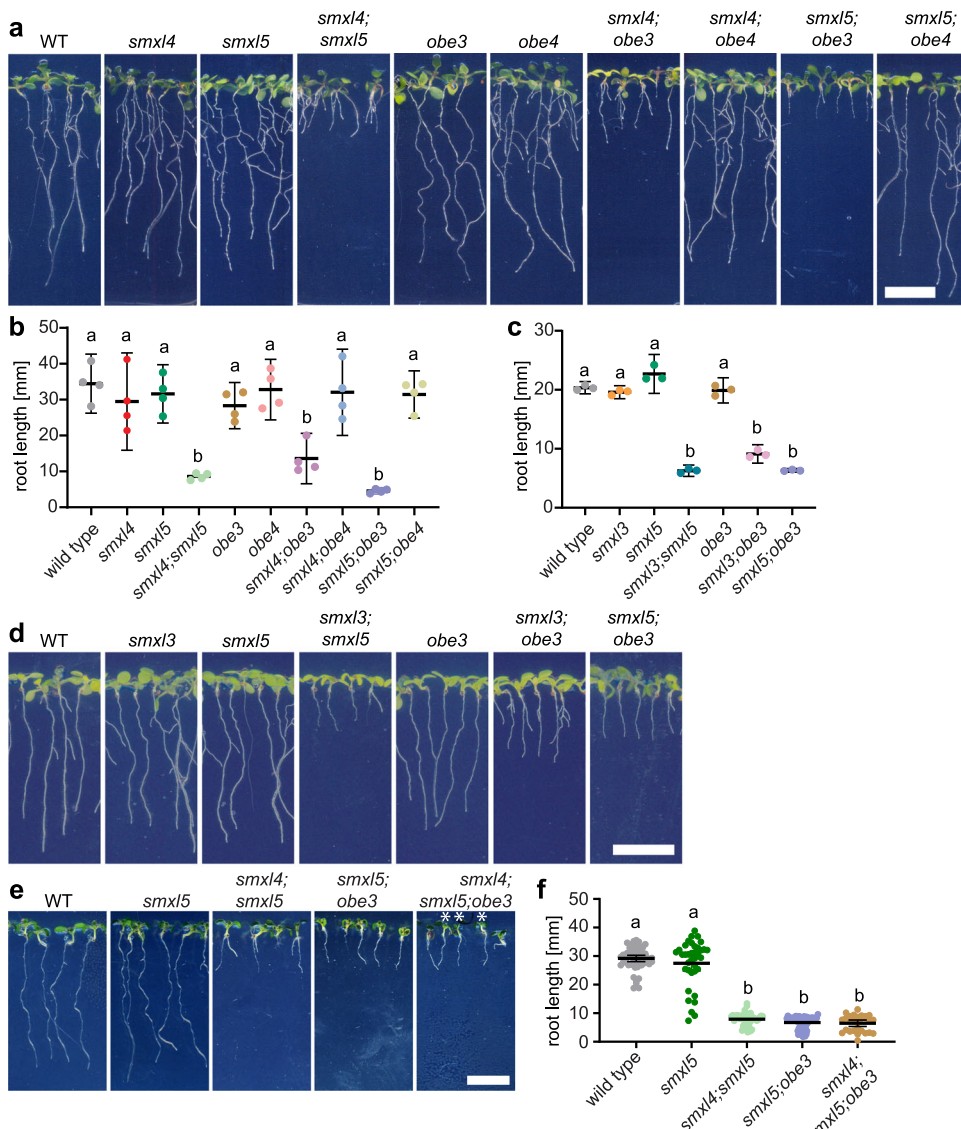

**Fig. 5 | OBE3 genetically interacts with SMXL3/4/5 in root growth regulation.**
**a** Ten-day-old wild-type, *smxl4, smxl5, smxl4;smxl5, obe3, obe4, smxl4;obe3, smxl4;obe4, smxl5;obe3, smxl5;obe4* seedlings are shown from left to right. Scale bar represents 1 cm. **b** Quantification of root length depicted in a. Mean values of four independent experiments (see Source Data File for exact sample numbers) were analyzed by a one-way ANOVA with post hoc Tukey HSD (95% CI). Statistical groups are marked by letters. **c** Quantification of root length depicted in d. Mean values of three independent experiments (see Source Data File for exact sample numbers) were analyzed by a one-way ANOVA with post hoc Tamhane-T2 (95% CI). Statistical groups are marked by letters. **d** Ten-day-old wild-type, *smxl3, smxl5, smxl3;smxl5,*

*obe3, smxl3;obe3, smxl5;obe3* seedlings are shown from left to right. Scale bar represents 1 cm. **e** Ten-day-old wild-type, *smxl5, smxl4;smxl5, smxl5;obe3,* and *smxl4;smxl5;obe3* seedlings are shown from left to right. Asterisks indicate *smxl4;smxl5;obe3* seedlings in a segregating *smxl4;smxl5;obe3*/+ population. Be aware that *smxl4;smxl5;obe3* triple mutants are lethal at later growth stages. Scale bar represents 1 cm. **f** Quantification of root length depicted in e. Error bars indicate standard deviation (see Source Data File for exact sample numbers). For determining statistical significance, a Wilcoxon/Kruskal–Wallis test was carried out followed by a Dunn post hoc analysis. Statistical groups are marked by letters.

of read alignment in *smxl5* phloem cells in comparison to non-phloem cells again including *OPS, BAM3, CVP2,* and *APL* (Fig. 8a, Supplementary Figs. 9 and 10, and Supplementary Data 2 and 3). In contrast, only 6 and 101 phloem-specific OCRs were detected in *smxl4;smxl5* and *smxl5;obe3* double mutants, respectively (Fig. 8a and Supplementary Data 2 and 3), suggesting a loss of phloem-specific chromatin signatures in both backgrounds. This impression was confirmed when plotting the relative abundance of reads mapping to the 1802 phloem-specific OCRs detected in wild-type in all the analyzed samples. In comparison to wild-type and *smxl5* mutants, *smxl4;smxl5* and *smxl5;obe3* double mutants showed a substantial reduction in the number of reads mapping to these regions (Fig. 8c).

This conclusion was also supported by direct comparison of the OCR profile obtained from GFP-positive samples. In total, only 65 OCRs showed significantly more or fewer aligned reads comparing *smxl5* mutants with wild-type (Fig. 8d and Supplementary Data 4, fold change >2, Poisson enrichment $P$ value < 0.05, using 40 M reads) arguing for a high similarity of the chromatin profile in phloem-related cells from both backgrounds. In contrast, *smxl4;smxl5* and *smxl5;obe3* double mutants differed considerably from both wild-type and *smxl5* mutants with regard to their phloem-related chromatin profile. *smxl4;smxl5* mutants showed 2963 OCRs in total with significant differences in read alignment in comparison to wild-type and, for *smxl5;obe3* mutants, 1882 differential OCRs were found. A similar

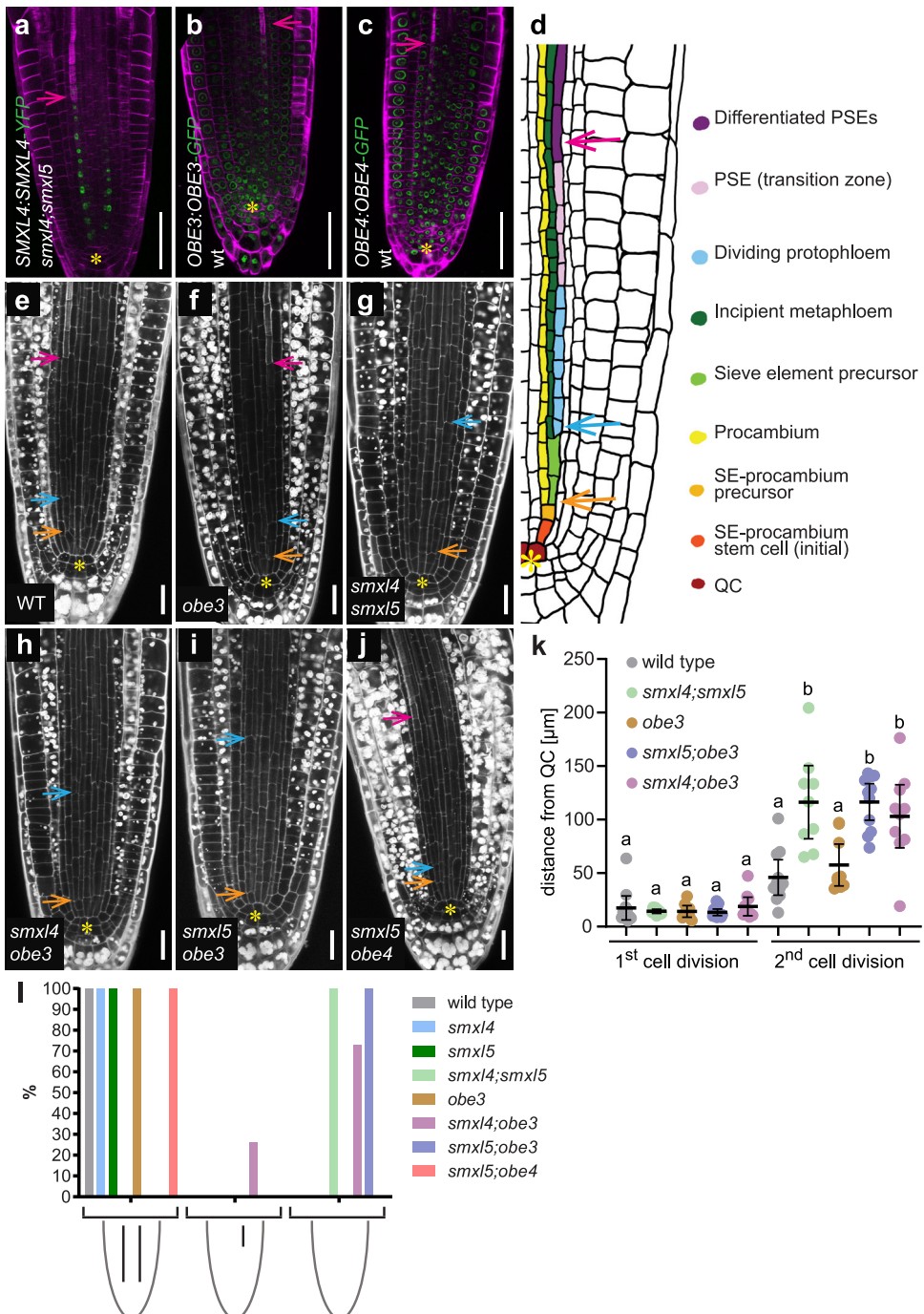

**Fig. 6 | *OBE3* interacts with *SMXL5* in protophloem formation. a–c** Phloem-specific activity of *SMXL4:SMXL4-YFP* (**a**) coincides with activity of *OBE3:OBE3-GFP* (**b**) and *OBE4:OBE4-GFP* (**c**) reporters in the developing phloem. Fluorescent signals (green) and cell wall staining by Direct Red 23 (magenta) were detected by confocal microscopy. Pink arrows point to the first differentiated SE indicated by enhanced cell wall staining. Scale bars represent 50 μm. **d** Schematic representation of one developing phloem pole at the root tip. Two periclinal cell divisions generate SE precursor and procambium (orange arrow) and proto- and metaphloem cell lineages (blue arrow), respectively. Differentiated SEs are marked by a pink arrow. The QC is marked by a yellow asterisk. Figure adapted from[61]. **e–j** Phloem formation in 2-day-old wild-type (**e**), *obe3* (**f**), *smxl4;smxl5* (**g**), *smxl4;obe3* (**h**), *smxl5;obe3* (**i**),

and *smxl5;obe4* (**j**) root tips. Cell walls were stained by mPS-PI (white). Yellow asterisks mark the QC. Enhanced mPS-PI staining indicates differentiation of SEs (pink arrows) in wild-type (**e**), *obe3* (**f**), and *obe4;smxl5* (**g**). Orange and blue arrows mark the first and second periclinal division, respectively, in the developing phloem cell lineage. Scale bars represent 20 μm. **k** The distance from the QC to the first and second periclinal division shown in (**d–i**) was quantified (see Source Data File for exact sample numbers). Statistical groups are indicated by letters and were determined by a one-way ANOVA with post hoc Tukey HSD (95% CI). Distances of 1st cell divisions and 2nd cell divisions were compared independently. **l** Phenotypic characterization of phloem development of 2-day-old wild-type, *smxl4*, *smxl5*, *smxl4;smxl5*, *obe3*, *smxl4;obe3*, *smxl5;obe3*, *smxl5;obe4* plants. *n* = 15 for each group.

situation was observed for both double mutants in comparison to *smxl5* mutants (Fig. 8d and Supplementary Data 4). Importantly, the number of differential OCRs was lower when comparing *smxl4;smxl5* and *smxl5;obe3* samples directly. In this case, only 421 differential

OCRs were detected (Fig. 8d), suggesting that both double mutants differed similarly from wild-type and *smxl5* mutants with regard to their chromatin profile in phloem-related cells. Beyond the overall OCR profile, the chromatin around phloem-related genes like *OPS*,

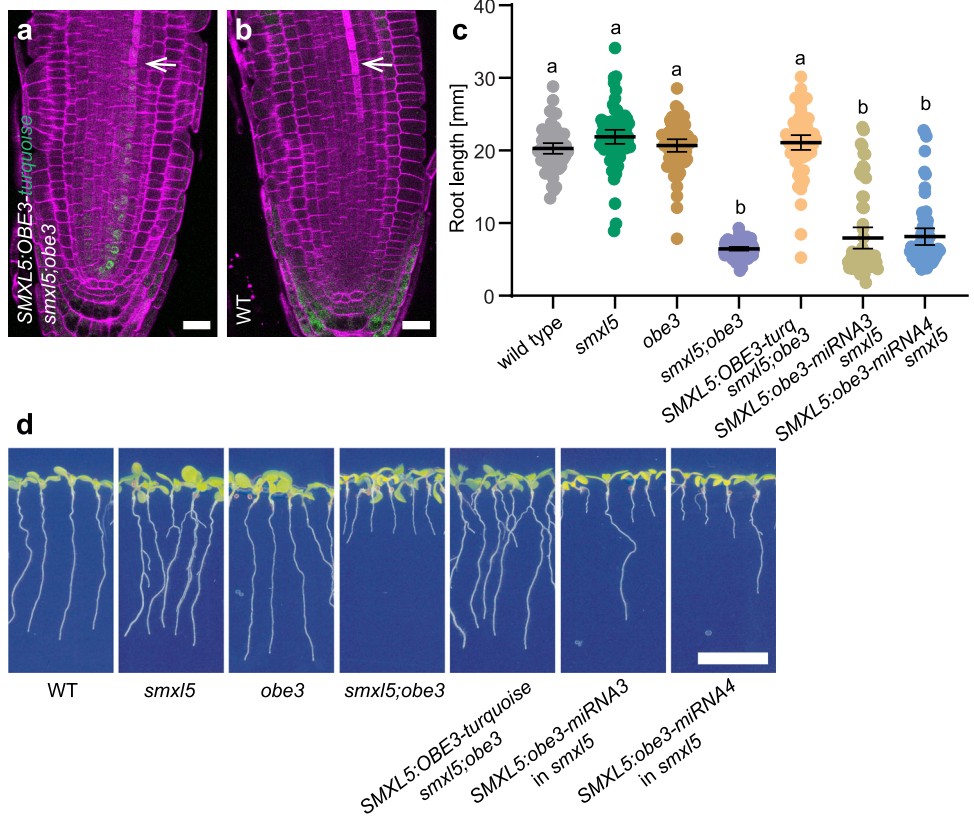

**Fig. 7 | Phloem-specific *OBE3* expression is sufficient for promoting root growth. a, b** Seven-day-old *smxl5;obe3* root tips carrying an *SMXL5:OBE3-turquoise* reporter (green signal, **a**) were compared to a non-transformed wild-type (WT) root tip (**b**). Cell walls were stained by DirectRed (magenta). White arrows mark differentiated SEs. Scale bars indicate 20 μm. Ten *SMXL5p:OBE3-turq;smxl5;obe3* plants and five wild-type plants were analyzed with similar results. **c** Root length quantification of plants depicted in (**d**). Results from one representative experiment out of three independent experiments (see Source Data File for exact sample numbers) are shown. Mean values were analyzed by one-way ANOVA with post hoc Tamhane-T2 (95% CI). Statistical groups are indicated by letters. **d** Ten-day-old wild-type, *smxl5, obe3, smxl5;obe3, SMXL5:OBE3-turquoise;smxl5;obe3, SMXL5:obe3-miRNA3;smxl5* and *SMXL5:obe3-miRNA4;smxl5* seedlings are shown from left to right. Scale bar represents 1 cm.

*BAM3*, *CVP2*, and *APL* displayed a more condensed conformation in GFP-positive nuclei from *smxl4;smxl5* and *smxl5;obe3* and the difference between GFP-positive and -negative nuclei was less pronounced in these cases (Supplementary Figs. 9 and 10).

When comparing the OCR profile obtained from GFP-negative samples, the difference between the genetic backgrounds was substantially lower with a maximum of 282 differential OCRs comparing wild-type and *smxl5* with the *smxl5;obe3* background, respectively (Fig. 8e and Supplementary Data 5). Together, these comparisons suggested a considerable and mostly phloem-specific difference between chromatin conformation in phloem-related cells when comparing wild-type and *smxl5* mutants on the one side and *smxl4;smxl5* and *smxl5;obe3* double mutants on the other side. Moreover, we concluded that phenotypic changes are similar in *smxl4;smxl5* and *smxl5;obe3* mutants, underlining the functional relatedness of *SMXL* and *OBE3* genes.

To see how chromatin conformation differed overall between the different genotypes, we defined a group of "phloem" genes (combined SUC2, APL and S32 domain genes according to ref. 45, Supplementary Data 6) and "non-phloem" genes (combined M1000, cortex, COBL9, GL2, AGL42, PET111, and LRC domain genes according to ref. 45, Supplementary Data 6). When analyzing chromatin conformation associated with the different gene groups, a similar pattern in wild-type and *smxl5* mutants was observed with a more open chromatin around transcriptional start sites (TSS) of non-phloem genes in GFP-negative samples compared to GFP-positive samples (Fig. 8b). Interestingly, in GFP-positive samples from *smxl4;smxl5* and *smxl5;obe3*

double mutants, overall chromatin conformation of phloem genes was similar to wild-type. In contrast, chromatin of non-phloem genes was more open around TSSs in GFP-positive samples and the conformation resembled the pattern in GFP-positive samples (Fig. 8b). These observations again indicated that phloem-specific chromatin signatures were equally reduced in phloem-related cells of *smxl4;smxl5* and *smxl5;obe3* double mutants reflecting a comparable function of both SMXL4/5 and OBE3 proteins. In contrast, when performing the same analysis for xylem-related genes (combined S4 and S18 domain genes according to ref. 45, Supplementary Data 6), no differential chromatin conformation could be detected comparing GFP-positive and GFP-negative samples or comparing the different genetic backgrounds (Supplementary Fig. 11). This indicated that there was no particular alteration of chromatin signatures around genes associated with other vascular tissues in *smxl4;smxl5* and *smxl5;obe3* double mutants arguing for a specific role of SMXL5 and OBE3 in phloem cells and in targeting phloem-related genes.

## Discussion
Cell-type specification is fundamental for establishing multicellular organisms and, in recent years, the phloem has become an instructive model for studying this aspect in plants[10,12,46,47]. With our study, we provide new insights into the regulation of (proto)phloem formation by revealing a role of the putative chromatin remodeling protein OBE3[31] and a direct interaction between the OBE3 protein and the central phloem regulator SMXL5. Based on our findings, we propose that both proteins

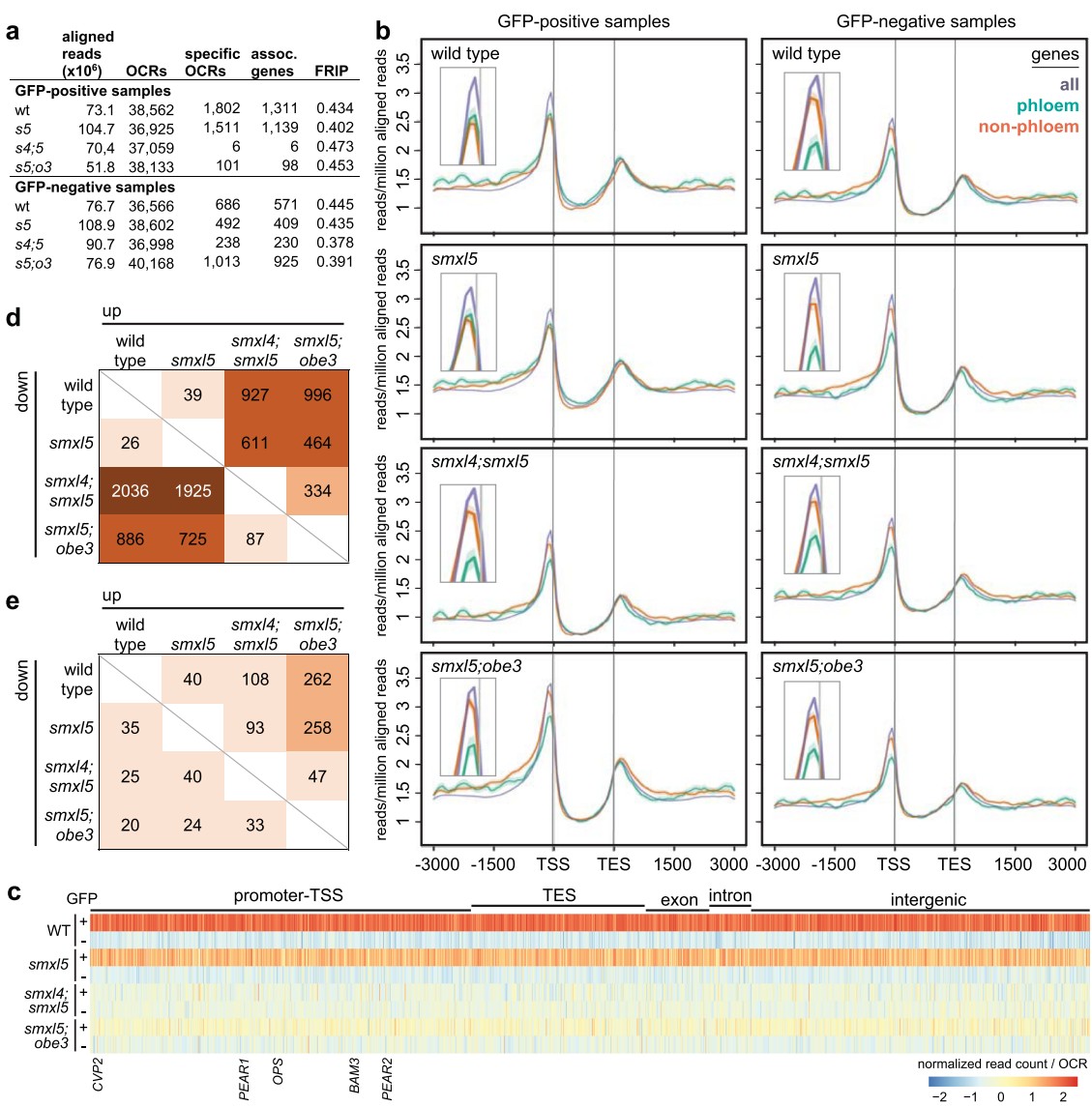

**Fig. 8 | *SMXL* and *OBE3* genes determine phloem-related chromatin profile.**
**a** Summary of ATAC-seq results for the different samples. Specific OCRs were identified by comparing GFP-positive and GFP-negative samples of the respective genetic backgrounds. After initial alignment, OCR detection and other subsequent analyses were conducted using 40 M reads aligning to the nuclear genome excluding duplicates for each sample. **b** Chromatin conformation profiles for all genes found in the Arabidopsis genome or only for phloem and non-vascular genes, according to ref. 45. Read alignment was adjusted to transcriptional start sites (TSS) and transcriptional end sites (TES). X-axes show distance to TSSs and TES in base-pairs. Inserts in all graphs are enlargements of the peaks identified close to TSSs.

**c** Heatmap showing the relative abundance of reads mapped to 1802 phloem-specific OCRs detected in wild-type (WT) in the different ATAC-seq samples. Read counts were normalized for each OCR and displayed according to the color code shown at the bottom. OCRs are moreover classified according to the relative position of reads to different functional regions. Reads mapped to the region −1 kb to +100 bp relative to the TSS, and −100 bp to +1 kb relative to the TES were classfied as promoter-TSS and TES, respectively. **d**, **e** Comparison of OCR profiles found in GFP-positive (**c**) and GFP-negative (**d**) samples. Numbers indicate the number of OCRs with significantly more ("up") or less ("down") aligned reads.

fulfill their role in phloem formation by promoting a distinct nuclear signature important for driving phloem development.

SMXL3/4/5 and OBE3 proteins are already expressed in phloem stem cells[5], which was so far not described for other phloem regulators[47]. Here, we detected OPS and BRX protein accumulation already in those stem cells raising the question of functional interdependence. The positive effect of *SMXL4* and *SMXL5* gene functions on OPS and BRX protein accumulation in those and more mature phloem cells suggests that *SMXL* genes are required for the establishment of a phloem-specific developmental program, including *OPS* and *BRX* gene activities. The different subcellular localization of OPS, BRX and BAM3 proteins on the one side and SMXL proteins on the other side argues against a more interconnected mechanism of both groups of regulators. The conclusion that both groups act on different aspects

of phloem formation is furthermore supported by our genetic analyses which revealed a combination of distinct phloem defects in respective double mutants. Here, we propose that SMXL proteins fulfill their role in the nucleus of phloem stem cells and beyond by direct interaction with OBE3. In fact, the *SMXL3/4/5-OBE3* interaction seems to be a prerequisite for both the initiation of phloem cell fate and a timely onset of differentiation. On the genetic level, *smxl4;obe3*, *smxl5;obe3* and the *smxl4;smxl5* double mutants share the same phloem defects meaning that they are deprived of protophloem formation within the RAM. Other reported phloem mutants have either problems with completing phloem differentiation in general, which is the case in mutants of the *APL* gene[17,48,49], or they develop "gap cells" in which SE differentiation is disturbed as in *ops* or *brx* single or in *cup2;cup2-like1* (*cvl1*) double mutants[9,11,15,16,35]. In contrast, *smxl5;obe3*, *smxl4;obe3*, and

multiple mutants of the *SMXL3/4/5* genes show the absence of all morphological hallmarks of phloem formation within the RAM. This suggests that *SMXL3/4/5* and *OBE3* together establish of a phloem-specific developmental program. Until recently, complete suppression of protophloem formation was so far only reported for roots treated with certain CLE peptides, such as CLE45, which signal through the leucine-rich repeat receptor-like kinase BAM3 and the pseudokinase CORYNE (CRN)[9,14]. However, the formation of "gap cells" in *ops* or *brx* mutants is suppressed in *BAM3*-deficient backgrounds[4]. This shows that none of those factors is required to obtain protophloem cell identity and proper differentiation in the first place[4,9]. Moreover, the epistatic relationship between *SMXL4/5* and *BAM3* genes as observed in this study shows that the SMXL and BAM3 pathways are functionally independent. Interestingly, a recent study showed that multiple mutants in phloem-associated DNA-BINDING WITH ONE FINGER (DOF) transcription factors show, similar to multiple *smxl* mutants, a fundamental defect in phloem formation[50]. In contrast to the *SMXL5* gene, ectopic *DOF* expression is sufficient to induce phloem formation[50] which, together with the observation that the *SMXL3* gene is a direct target of the same DOFs[51], suggests that *SMXL* genes are among the essential targets of phloem-associated DOF transcription factors to induce the whole of a phloem-specific developmental program.

PHD-finger motifs as carried by the OBE3 protein are known to be epigenetic readers binding to histone H3 tails carrying distinct post-translational modifications such as trimethylation of lysine 4 (H3K4me3) or lysine 9 (H3K9me) marking actively transcribed or silent chromatin regions, respectively[33]. Although PHD-finger proteins themselves are not necessarily activating or repressing, they can indirectly modify transcription by recruiting chromatin-modifying complexes[52]. Indeed, OBE proteins have been proposed to remodel chromatin structure during embryogenesis and thereby transcriptionally activate RAM initiation factors[31]. SMXL proteins share an ethylene-responsive element binding factor-associated amphiphilic repression (EAR) motif which interacts with transcriptional regulators of the TOPLESS (TPL) family[53] and they have recently shown to bind DNA directly and repress activity of target genes[25]. Indeed, the strigolactone signaling mediators SMXL6, SMXL7, SMXL8 and DWARF53 (D53), a SMXL protein from rice, directly interact with TPL-like proteins[22,24,26]. TPLs are transcriptional co-repressor which can recruit histone deacetylases (HDAC) and, thereby, induce chromatin condensation and transcriptional suppression[26,54]. In addition to EAR motifs, SMXLs share a conserved double caseinolytic protease (Clp) domain with ATPase activity that resembles heat shock protein 101 (HSP101)[23,55]. As recently proposed, the p-loop ATPase domain of D53 fosters the formation of TPL hexamers and the threading of DNA through a central pore of this hexamer inducing nucleosome repositioning and/or higher-order chromatin reorganization[26]. Because the mechanistic relationship between activating OBE proteins and repressing SMXL proteins is currently unclear, it is difficult to speculate about the role of an OBE/SMXL protein complex with regard to its direct effect on chromatin structure. We observed that non-phloem genes show a more open conformation in cells in which the *SMXL5* promoter is active but that chromatin around phloem-related genes is more condensed. This makes it possible that such a complex holds activating or repressing activity depending on which genes are directly targeted. The identification of these direct targets and the detailed analysis of the effect of SMXL/OBE complex(es) on their activity and chromatin conformation will be required to decide between these possibilities.

## Methods

### Plant material

Genotypes of plant species *Arabidopsis thaliana* (L.) Heynh. of the ecotype Columbia (Col) used for genetic analysis are listed in Supplementary Table 2. Sterile seeds were stratified in microcentrifuge tubes containing dH$_2$O at 4 °C in the dark for 3 days and then sown in rows on ½ Murashige and Skoog (MS) medium plates supplemented with 1% sucrose and grown vertically. Seedlings were grown in long day (LD, 16 h light and 8 h dark) conditions at 21 °C for 2–10 days. Seeds were liquid sterilized by 70% ethanol supplemented with 0.2% Tween-20 for 15 min, washed twice with 100% ethanol and air dried under sterile conditions. *bam3-2* (SALK_044433), *obe1-1* (SALK_075710), *obe3-2* (SALK_042597) and *obe4-1* (SALK_082338) mutants were obtained from the NASC stock center. *smxl3-1* (SALK_024706), *smxl4-1* (SALK_037136), *smxl5-1* (SALK_018522), and *obe2-2* mutants were described before[5,32]. *smxl4-1;smxl5-1;bam3-3* triple mutants were generated by CRISPR/Cas9 targeting *BAM3* in a *smxl4-1;smxl5-1* mutant background (creating the *bam3-3* allele), leading to a G insertion after position 707 of the CDS and to the generation of a stop codon after 255 aa of the BAM3 protein (i.e., in the LRR receptor domain). All other lines are referenced in the text.

### Yeast-two-hybrid

The yeast-based screen for proteins interacting with SMXL5 was performed by Hybrigenics (Evry, France)[38]. The yeast strain *AH109* was used for the yeast-two-hybrid assay according to Matchmaker™ Two-Hybrid System 3 (Clontech, Palo Alto) and grown on YPD (full medium) or SD (selective drop-out medium)-agar plates for 3–5 days at 28 °C, then stored at 4 °C and stroked onto new plates every 10 days. Dilution series (OD$_{600}$ 1-0.001) of transformed yeast strains were grown for 3 days on selective drop-out medium (SD) -Leu/-Trp/-His/-Ade selecting for protein interaction or SD -Leu/-Trp selecting for the presence of the plasmids. When grown in liquid YPD medium for transformation, yeast was grown overnight at 28 °C with shaking at 250 rpm. For expressing the GAL4BD-SMXL5 fusion protein in yeast the open reading frame of the *SMXL5* gene was cloned into XmaI/BamHI sites of the *pGBKT7* plasmid (Clontech, Palo Alto) resulting in *pEW6*. For expressing the GAL4AD-OBE3 fusion protein, the open reading frame of the *OBE3* gene was cloned into XmaI/BamHI sites of the *pGADT7* plasmid (Clontech, Palo Alto) resulting in *pEW11*.

***Agrobacterium tumefaciens***. The *Agrobacterium tumefaciens* genotypes C58C1: RifR with pSoup plasmid (TetR) or ASE: KanR, CamR with pSoup+ plasmid (TetR) were used for transformation of *Arabidopsis thaliana* or infiltration of *N. benthamiana* leaves and grown at 28 °C overnight in liquid YEB medium on a shaker (180 rpm to an OD$_{600}$ > 1) or plated on YEB-plates and grown in an incubator[56–58]. Antibiotics were used for plasmid selection, as listed in Supplementary Table 2.

### Transformation of *Arabidopsis thaliana*

Arabidopsis plants were transformed by the floral dip method[59]. To do so, 150 ml of Agrobacteria culture carrying the respective plasmid were pelleted at 3500 × *g* for 15 min at RT and the pellet was quickly rinsed by tap water. The bacteria pellet was subsequently dispersed in 100 ml tap water with 5% sucrose, and 20 μl of detergent SILWEET-L77 were added. Flowers and buds of Arabidopsis plants were hung into the bacterial dispension for 5 min and subsequently incubated at room temperature overnight in the dark before being put back into the growth chamber. Transformed seeds were harvested after 3 weeks and T1 plants were selected on ½ MS plates supplemented with the respective plant resistance listed in Supplementary File 2.

### Root length measurements

For measuring root lengths, seedlings were scanned by a commercial scanner and analyzed using ImageJ 1.49d[60]. For CLE45 treatments, plants were germinated on normal MS media (mock) or media containing 50 nM CLE45 and root length measurements were performed after 5 days.

## Genotyping

Genotyping was performed by PCR using primers listed in Supplementary Table 1. Further information about standard DNA extraction and genotyping can be found in ref. [61].

## Transient protein expression in *N. benthamiana*

*N. benthamiana* plants were used for transient protein expression and grown in the greenhouse at ~25 °C and watered daily. Transformed *Agrobacteria* were stored as glycerol stocks and grown in a 10 ml YEB liquid culture prior to use. The densely grown culture was centrifuged at $3500 \times g$ for 5 min at RT. The supernatant was removed and the pellet was washed with 5 ml induction buffer and re-suspended in 10 ml induction buffer. Culture densities were adjusted to an $OD_{600}$ of 1.0. Prior to infiltration, these bacterial solutions were mixed with *Agrobacteria* expressing *35S:P19* in a ratio 1:2 and incubated in the dark for 2–3 h[62,63]. *N. benthamiana* leaves were infiltrated with the mixtures using a 1-ml syringe (Becton Dickinson S.A., Heidelberg, Germany). Leaves were harvested 3 days after infiltration.

## Protein extraction, immunoprecipitation, and western blot

Infiltrated *N. benthamiana* leaves were frozen in liquid nitrogen and ground by a mortar. Proteins were extracted by mixing the leaf powder 1:1 with extraction buffer (50 mM $Na_3PO_4$, 150 mM NaCl, 10% glycerol, 5 mM EDTA, 10 mM β-mercaptoethanol, 0.1% triton X-100, 2 mM $NaVO_4$, 2 mM NaF, 20 μM MG-132, 1 mM PMSF, 1× cOmplete™ Protease Inhibitor Cocktail (Roche; Basel, Switzerland)). Each sample was vortexed for 10 s and centrifuged at $17000 \times g$ for 10 min at 4 °C. The protein extract was retrieved by sieving it through a nylon mesh. Protein quantities were measured by Bradford assays according to the manual provided with the Bio-Rad Protein Assay Dye Reagent Concentrate (Bio-Rad Laboratories; Hercules, USA). Proteins were immunoprecipitated by 50 μl Anti-HA MicroBeads (Miltenyi Biotec, Bergisch Gladbach, Germany) after incubation for 2.5 h at 4 °C while slowly rotating. Beads were captured by μ Columns (Miltenyi Biotec, Bergisch Gladbach, Germany) on magnetic stands by following the user manual and washed three times by 200 μl Wash buffer I (extraction buffer without β-mercaptoethanol) and two times by 200 μl Wash buffer II (50 mM $Na_3PO_4$, 150 mM NaCl, 10% glycerol, 5 mM EDTA). Proteins were eluted by 2x Laemmli buffer (95 °C) and separated by size on a SDS-PAGE with subsequent western blotting. Detailed procedures can be found in ref. [61]. SMXL5-3xHA and 6xMyc-OBE3 bands were detected by antibodies Anti-HA-Peroxidase High Affinity (3F10) (Roche; Basel, Switzerland) or c-Myc Antibody (9E10) sc-40 HRP (Santa Cruz Biotechnology, Santa Cruz, USA), respectively and visualized by chemiluminescence agents SuperSignal™ West Femto Maximum Sensitivity Substrate (Thermo-Scientific; Waltham, USA) by an Advanced Fluorescence and ECL Imager (Intas Science Imaging Instruments, Göttingen, Germany).

## FRET-FLIM analyses

FRET-FLIM analyses[64,65] were performed using a Leica TCS SP8 microscope (Leica Microsystems, Germany) equipped with a rapidFLIM unit (PicoQuant). Images were acquired using a ×63/1.20 water immersion objective. For the excitation and emission of fluorescent proteins, the following settings were used: mGFP at excitation 488 nm and emission 500–550 nm; and mCherry at excitation 561 nm and emission 600–650 nm. The lifetime τ [ns] of either the donor-only expressing cells or the cells expressing the indicated combinations was measured with a pulsed laser at an excitation light source of 470 nm and a repetition rate of 40 MHz (PicoQuant Sepia Multichannel Picosecond Diode Laser, PicoQuant Timeharp 260 TCSPC Module and Picosecond Event Timer). The acquisition was performed until 500 photons in the brightest pixel were reached. To obtain the GFP fluorescence lifetime, data processing was performed with SymPhoTime software and bi-exponential curve fitting and a correction for the instrument response

function. Statistical analysis was carried out using the JMP 14 software (JMP, USA).

## Direct Red 23 staining

To preserve fluorescent signals in roots, seedlings were fixed in a vacuum chamber for 1 h by 4% (w/v) PFA dissolved in PBS. The tissue was washed twice by PBS and cleared with ClearSee solution for a minimum of 2 days, according to ref. [64]. Cleared seedlings were stained by 0.01% (w/v) Direct Red 23 in ClearSee solution for 1 h. Excess staining was removed by clearing once again in pure ClearSee solution for 1 h.

## mPS-PI staining

Whole 2-day-old seedlings were submerged in fixative (50% methanol and 10% acetic acid) for 24–96 h at 4 °C. After rinsing seedlings with sterile water, they were incubated in 1% periodic acid at room temperature for 40 min. After rinsing again with sterile water, roots were stained in Schiff reagent with propidium iodide (100 mM sodium metabisulphite and 0.15 N HCl with freshly added propidium iodide to a final concentration of 100 μg/mL) for 2–3 h or until plants were visibly stained. Stained roots were quickly transferred onto objective slides and covered with few drops of chloral hydrate solution (4 g chloral hydrate, 1 mL glycerol, and 2 mL water). The samples were kept at room temperature in sealed plastic boxes with wet paper towels to prevent drying out overnight. After 16 h, excess chloral hydrate was removed from the samples and roots were mounted in Hoyer's solution (30 g gum arabic, 200 g chloral hydrate, 20 g glycerol, and 50 mL water) and gently covered with a cove slip. After drying the slides for at least 3 days at room temperature in the dark, PI-stained tissues were excited at 561 nm (DPSS laser) and detected at 590–690 nm using a Leica SP5[49].

## Confocal microscopy

For confocal microscopy, TCS SP5 or SP8 microscopes (Leica Microsystems; Mannheim, Germany) were used. GFP signals were excited by an argon laser at 488 nm, collecting the emission between 500–575 nm. YFP was excited by an argon laser at 514 nm and the emission detected in a range of 520–540 nm. DirectRed stained tissue was excited at 561 nm (DPSS laser) and emission was detected at wavelengths >660 nm. mPS-PI-stained tissue was excited at 561 nm (DPSS laser) and emission was detected at 590–690 nm.

## Molecular cloning and miRNA generation

*OPS:SMXL5-VENUS* (*pNT52*), *OPS:ER-VENUS* (*pNT53*), *BAM3:SMXL5-VENUS* (*pNT49*), *BAM3:ER-VENUS* (*pNT50*), *CVP2:SMXL5-VENUS* (*pNT16*), *CVP2:ER-VENUS* (*pNT69*), *APL:SMXL5-VENUS* (*pNT10*), *APL:ER-VENUS* (*pNT68*), *SMXL4:BRX-VENUS* (*pNT72*), *SMXL5:OBE3-turquoise* (*pEW72*), *35S:5xc-Myc-OBE3* (*pEW78*),*35S:SMXL5-mCherry* (*pVL122*), *35S:OBE3-mGFP* (*pVL127*), *35S:mCherry-NLS* (*pMG103*) and *UBI10:mGFP-mCherry-NLS* (*pCW194*) constructs were generated by using appropriate modules following the GreenGate procedure[65]. Destination modules, entry modules, and correlating primers for amplifying DNA fragments for generating entry modules are depicted in Supplementary Table 1. In case reporter proteins were targeted to the endoplasmatic reticulum (ER), they were fused to the appropriate motifs[66]. *miRNAs* targeting *OBE3* transcripts were designed and cloned according to the manual provided by the WMD3—Web MicroRNA Designer Version 3 (Max Planck Institute for Developmental Biology, Tübingen. http://www.weigelworld.org) with primers listed in Supplementary Table 1. To generate *35S:SMXL5-3xHA* (*pEW33*), ssDNA sequences coding for 3xHA (Supplementary Table 1) were annealed by gradual cool-down from 80 °C to 50 °C and inserted into the vector *pGreen0229:35S*[57] using BamHI/XmaI sites, resulting in plasmid *pEW31*. Next, the *SMXL5* CDS was amplified by primers listed in Supplementary Table 1 and cloned into BamHI/XbaI sites of *pEW31* resulting in *pEW33*. For generating the *35S:SMXL5-YFP* (*pKG34*) construct, the *SMXL5* CDS was amplified by

primers listed in Supplementary Table 1 and cloned into BamHI/XbaI sites of *pGreenO229:35S* resulting in *pKG33*. Next the YFP fragement was amplified by primers again listed in Supplementary Table 1 and cloned into BamHI/XmaI sites. Further information about detailed cloning procedures can be found in ref. 61.

## Fluorescence-activated nucleus sorting (FANS) and ATAC-seq

Nucleus extraction for FANS/ATAC-seq[67] was carried out on ice. Approximately 1 g of 3-week-old Arabidopsis seedlings grown in ½ MS LD conditions were collected in a 60-mm Petri dish and thoroughly chopped using a razor blade for 5 min in nucleus isolation buffer (1× NIB, CelLytic™-PN Isolation/Extraction buffer, Sigma-Aldrich, cat.no. CELLYTPN1) supplemented with 10 µg/ml Hoechst 33342 (Sigma-Aldrich, B2261) as the final concentration. After incubation for 15 min at 4 °C in darkness, the nucleus suspension was applied to 50-µm nylon strainers mounted into a 30-µm nylon strainers at the top (Sysmex, CellTrics), and filtered further by a FACS tube with 35-µm strainer cap (Corning, #352235). 15,000 nuclei for each sample were then sorted according to their GFP signal levels into a 300 µL collection buffer (15 mM TRIS-HCl pH 7.5, 20 mM NaCl, 80 mM KCl, 0.1% Triton) by a BD FACSAriaTM IIIu cell sorter using a 100 µm sort nozzle and a 30 kHz drop drive frequency. The gate for GFP + nuclei was set using wild-type as a reference (Supplementary Fig. 7b). Next, samples were centrifuged at $3000 \times g$ for 10 min at 4 °C. The supernatant was partially removed, and the nuclei were re-suspended in 300 µL Tris-Mg buffer (10 mM Tris-HCl pH 8, 5 mM $MgCl_2$). A second washing step was performed using centrifugation at $1000 \times g$ for 10 min at 4 °C and the supernatant was removed completely. For tagmentation, the TDE1 (Nextera Tn5 Transposase) Tagment DNA Enzyme (Illumina kit #20034198) was immediately applied to the samples by resuspending the nuclei in the transposition reaction mix and the suspension was incubated for 30 min at 37 °C while gently shaking[42]. The samples were then purified (NEB, Monarch Nucleic Acid Purification Kit #20034198), and 12 µl of the transposed DNA were used for amplification by PCR (8 µl $H_2O$, 2.5 µl 25 µM Customr Nextera PCR primer 1[68], 2.5 µl 25 µM Custom Nextera PCR Primer 2 and 25 µl NEB Next High Fidelity 2x PCR Master Mix (NEB, #M0541) with 1 cycle of (72 °C for 5 min, 98 °C for 30 s) and 16 cycles of (98 °C for 10 s, 63 °C for 30 s, 72 °C for 1 min). The library was cleaned up using the Coulter Agencourt AMPure XP kit (Beckman-Coulter, #10136224) applying a 1:1 vol:vol (PCR product:beads) and eluted in 15 µl 15 mM Tris. The library was sequenced using NextSeq 550 (Illumina) using High-Output with 40 cycles Paired End mode.

## Analysis of sequencing data

Reads were mapped to Arabidopsis TAIR 10 genome by bowtie2 (v2.2.6)[69]. Output sam files were converted to bam files and then trimed with "-q 2" and "1 2 3 4 5" parameters in samtools (v0.1.19)[70] to remove mitochondrial and chloroplast sequence. Output bam files were treated with picard MarkDuplicates (v2.25.5) to remove duplicated reads. Then, output files were passed through samtools view again, to obtain 40 M reads (targeted) for further analysis. For the profiles, bigWig files were generated by bamCoverage in deepTools (v3.5.1)[71] from bam files with 10 bp bin size, and visualized by igv[72]. For peak calling, bam files were converted to TagDirectory by homer (v4.10.3)[73], and peaks were called by findPeaks in homer with "-region -minDist 150" option. The fraction of reads in called peak regions (FRiP values) were calculated using "intersect -wa" function in bedtools (v2.27.1)[74]. TSS profiles were made by ngs.plot (v2.47)[75]. To obtain differential peaks, first, the peak regions of all samples were merged using the mergePeaks function in homer and the reads mapped on the merged peak regions were kept by "bedtools intersect -wa" for each sample. Then, the pairwise differential peaks were obtained by getDifferentialPeaks function in homer with "-F 2.0 -P 0.05" option with the standard normalization to the number of reads of each sample. By using the trimmed reads mapped on the merged peak regions, different signal-to-noise ratios among samples were also normalized. Reads mapped to the phloem-specific OCRs in wild-type were counted by multicov function in bedtools, and heatmap was generated using pheatmap in R using the "scale = 'row' " option.

Although several replicates have been processed, only the result of one of those is included in this manuscript. This decision was made based on the observation that the number of identified OCRs comparing phloem and non-phloem tissues was considerably lower in the other replicates (e.g., 1488 phloem-specific OCRs in the included replicate in wild-type vs. 116 phloem-specific OCRs in an independent replicate) and, thus, the sensitivity was substantially lower. We would argue, though, that the provided dataset is reliable. The main reason for this conclusion is that the dataset contains internal controls demonstrating reliability of the data and allowing the estimation of technical noise. The similarity of wild-type and *smxl5* mutants which do not show phloem defects with regard to their phloem- and non-phloem-related chromatin profiles argues for a technically solid and biological relevant analysis. The same is suggested by the similarity of chromatin profiles of the *smxl4;smxl5* and *smxl5;obe3* mutants which show similar phloem defects. Also, the differential profile of phloem-associated genes in wild-type and *smxl5* mutants and the reduction of these differences in the two double mutants support this conclusion. With GFP-positive and GFP-negative samples for each of the four genotypes, we analyzed eight samples by ATAC-seq in total which, although not being replicates in a strict sense, confirm each other and, as we think, are therefore a valuable source of information.

## Quantification and statistical analyses

Statistical analyses were performed using IBM SPSS Statistics for Windows, Version 22.0. Armonk, NY: IBM Corp or using GraphPad Prism version 6.01 (GraphPad Software, La Jolla, USA). Means were calculated from measurements with sample sizes as indicated in the respective figure legends. In general, all displayed data represents at least three independent, technical repetitions, unlike otherwise indicated. Error bars represent ± standard deviation. All analyzed datasets were prior tested for homogeneity of variances by the Levene statistic. One-way ANOVA was performed, using a confidence interval (CI) of 95% and a post hoc Tukey HSD for comparisons of five or more datasets of homogenous variances or a post hoc Tamhane-T2 in case variances were not homogenous. Graphs were generated in GraphPad Prism version 6.01 or 9.2 (GraphPad Software, La Jolla, USA) or in Excel (Microsoft, Redmond, USA).

## Reporting summary

Further information on research design is available in the Nature Portfolio Reporting Summary linked to this article.

## Data availability

Raw ATAC-seq data were deposited to the NCBI GEO archive[76] under the accession GSE184344. Oligonucleotides, Accession codes and unique identifiers of used material are mentioned in Supplementary Tables and "Methods". All used material is available upon request to the corresponding author. Source data are provided with this paper.

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

## Acknowledgements

This work was supported by the SFB 1101 to K.H., J.U.L, T.G., and SFB 873 (Deutsche Forschungsgemeinschaft, DFG) to J.U.L. and T.G., an ERC Consolidator grant (PLANTSTEMS, #647148) to T.G., a postdoctoral fellowship of the Japan Society for the Promotion of Science (JSPS Overseas Research Fellowships 201960008) and a JST PRESTO grant [JPMJPR2046] to D.S., and a PhD student fellowship of the Cusanuswerk to N.T. We are grateful to Christian Hardtke (University of Lausanne, Switzerland), Dolf Weijers, Shunsuke Saiga (both at Wageningen University, The Netherlands), Andy Maule (John Innes Centre, UK), Sebastain Wolf and Karin Schumacher (both Heidelberg University, Germany) for providing seed material and constructs. We are also grateful to Monika Langlotz (Cell Networks Flow Cytometry & FACS Core Facility, Heidelberg University, Germany) and David Ibberson (Cell Networks Deep Sequencing Core Facility, Heidelberg University, Germany) for providing excellent technical support. For the publication fee we acknowledge financial support by the DFG within the funding programme „Open Access Publikationskosten" as well as by Heidelberg University.

## Author contributions

Conceived and designed the experiments: E.W., N.T., F.W., V.L., D.S., L.L., K.H., and T.G. Performed experiments: E.W., N.T., F.W., L.L., I.J., V.L., P.H., Y.X., M.G., and C.W. Analyzed the data: E.W., N.T., F.W., K.H., J.U.L., D.S., and T.G. Wrote the paper: E.W. and T.G.

## Funding

## Competing interests

The authors declare no competing interests.
