## [Peer Review File · Nature Communications]

OBERON3 and SUPPRESSOR OF MAX2 1-LIKE proteins form a regulatory module driving phloem developmentReviewer #1 (Remarks to the Author):

The manual by Wallner et al revealed OBE3 and SMXL5 could interact with each other in nuclei of phloem stem cells and are critical to establish a phloem-specific chromatin profile. They have provided abundant evidences of genetic interaction and protein interaction, and also performed ATAC-seq to illustrate the chromatin accessibility alterations. Overall, this is a comprehensive work to understand the critical roles of OBE3 and SMXLs in the spatial specificity of cell fate decisions. I think the story is likely to be of interest to plant biologists in root developments. However, I have a few concerns, especially on the ATAC-seq parts.

For the ATAC-seq part:

1. Since the authors used SMXL5:H4-GFP as a marker to isolate phloem-associated cells, the SMXL5 promoter therefore are likely more accessible in the GFP+ cells than the GFP- cells. However, higher enrichments could be found in GFP+ samples of WT, *smxl5* and *smxl5;obe3*, but not in *smxl4;smxl5* (see attached figure), which suggested that either the material is not correct or the isolation of phloem-associated nuclei in *smxl4;smxl5* failed.
2. There're no replicates for ATAC-seq. The differences between samples were limited in arabidopsis. It's better to include replicates to estimate the influence of technical.
3. For the ATAC-seq analysis: (1) According to the genome coverage images and the bigwig files in GEO, the qualities of the ATAC-seq samples looked good. But it's better to calculate the FRiP values, since it can also reflect the signal enrichments of each sample. (2) Although the authors have picked up equal reads for the analysis, normalization are still required to deal with different signal/noise ratio. (3) The authors may quantify the chromatin accessibilities with reads depth in the putative differential peaks and put all samples together to make a heatmap, which would be helpful to classify the differential peaks and also easier for the readers to understand.
4. Chromatin accessibilities were highly associated with gene expression. If the authors could perform RNA-seq with the isolated nuclei, it will much better support the conclusions and also reveal the effects of chromatin variations on gene expressions.

Other points:

5. *obe4;smxl5* did not show root deficient phenotypes like *obe3;smxl5*, and OBE3 and OBE4 have similar expression patterns in root (Fig. 6B-6C). How about the protein interaction between OBE4 and SMXL? It may help to explain the functional differentiation between OBE3 and OBE4.
6. The localization of OBE3-mCFP in Fig. 4f looks different from those in Fig. 4c and 4m, it would be better to change to a more clear one.
7. Fig. 6b, please rotate the root direction to vertical.

Reviewer #1 Attachment on the following page

Genome coverage showing the chromatin accessibilities around *SMXL5*.

Reviewer #2 (Remarks to the Author):

In the manuscript by Wallner and co-workers, the authors elegantly show how SMXL4/5 act during early phloem development before other known regulators such as OPS, BRX, BAM3 etc. The authors convincingly show how SMXL4/5 as early regulators of phloem development interact with OBE3 in the nucleus. Both proteins overlap in expression domain in the early phloem cells, making it likely that the interaction indeed occurs here. Given the published function of OBERON proteins, the authors go on by undertaking chromatin profiling. Although these results nicely confirm their previous findings, I do feel that the authors are making strong conclusions based on this data (see comments below) and even extrapolate this to claiming that the SMXL4/5-OBE3 complex determines phloem cell fate. I have some reservations about these claims as in some cases, I see a correlation at best. Additionally, the genetic interaction studies are very clear and thoroughly performed, but the conclusions drawn are at moments a bit strong in my opinion as there are equally possible alternative hypothesis based on this data which were not explored. I want to make clear that I am not doubting the data nor the conclusions of the authors throughout the manuscript, but feel that a better argumentation and, at places, a more careful way of describing the conclusions would be needed to ensure the conclusions are in line with the experimental data.

Fig1F and H: what is the GFP signal in the epidermis/cortex cells?

Line 126: 'In contrast to...' I do not understand this as the previous sentence is about marker which are also reduced or absent in the smxl double mutant. So, why does this sentence start with 'in contrast' when the observation is the same.

Line 129: the first reference to Fig S1 starts with panel J. Could these be re-ordered to match the order in the text?

Line 131: the interaction between these components (BRX, OPS, BAM3, CLE45 etc.) is not that trivial to an outsider. Can you integrate a schematic of the pathway somewhere? This would make it easier to follow the genetic experiments that are done and place them in the correct framework. As one example, it is not directly clear why the experiments in the last paragraph related to BAM3 and smxl double mutant are included. Can the authors provide some more background and rationale please? Why were similar experiments not done for the other factors like OPS, BRX etc?

Fig1: It would be useful to have insets (enlargement) for the BAM and CVP2 reporters to show if it is expressed or not (similar to what is done for OPS and BRX.

Line 147: Can the authors explain why the root length in the rescue with the OPS and BAM promoter is significantly longer than WT? Is this over-compensation due to stronger expression compared to the SMXL promoters?

Line 156: what is the rationale for only testing the BRX protein in the complementation of the smxl double mutant? Why not OPS, BAM3 etc.?

Line 166: the genetic data presented in this paragraph is used at the end to conclude that 'SMXL5 and OPS/BRX genes play roles at different steps during phloem formation with SMXL5 acting upstream'. Can these conclusion be drawn from this genetic data? First, how does it show that SMXL5 acts upstream of OPS/BRX? Next, the fact that the combination of two mutants showing a partial or weak phenotype results in a stronger phenotype can be interpreted in several ways. For example: either they act in the same pathway and knock-out of both makes the overall phenotype worse; or they act in different pathways and the combinatorial effect of perturbing two (unrelated) pathways gives a stronger phenotype compared to the individual ones. Both options are valid in my opinion, making it very difficult to draw conclusions from this type of genetic data. Can the authors explain better how this data unambiguously shows that they work in different steps of phloem formation (based on this genetic data) and how this data

shows that SMXL5 acts upstream? Stronger evidence could be that e.g. overexpression of one factor has no effect when the downstream factor is removed. This would show more clearly that one acts upstream of the other.

Fig3B: would be much easier to interpret this figure if the % phenotypes were clustered per genotype instead of per phenotype. It now takes very long to understand this graph.

Fig 4A: what is the + and - control? This is not indicated in the figure or legend.

Fig 4B: for completeness, please show the entire blots in supplemental and a respective loading control.

Fig 4C: what are these nuclear subdomains and why are they perfectly spherical? If associated to e.g. certain genomic regions, I would expect them to be so spherical. Is there any reference in literature to this? Related, why is the circumference of the subdomain from panel C shown as a reduced expression ring in panel D? Is this a bleed-through issue?

Line 239: again in this case, the genetic data does not unambiguously show 'a concerted action of OBE3 and SMXL3, SMXL4 or SMXL5 genes during primary root growth' as combining partial phenotype mutants from unrelated pathways can also result in stronger phenotypes. Although I am not at all questioning the overall conclusions drawn, the data presented in e.g. Fig5 is not strong evidence for genetic interaction between these players in my opinion as the alternative hypothesis of increased phenotypes upon combination of unrelated pathways remains a viable option.

Fig6E-J: what are the massive amount of white dots staining in the outer cell layers being stained? I have seen this is much reduced amount in some mutant lines, but never in this extent and also in the WT roots. Is this a specific modification to the staining?

Fig 7B: why is there green staining in the lateral root cap if this is a WT plant? And if this is background fluorescence, then why do we not see this in panel A?

Line 375: How can a meta-analysis of chromatin profiles be proof for a comparable function of SMXL4/5 and OBE3 proteins? Related, the authors are using a helicopter-view chromatin profile analysis to draw conclusions about the fact that SMXL and OBE3 determine phloem-related chromatin profiles. I think this is taking the interpretation of the data one step to far. Although I do see small differences in e.g. Fig8B, it is not clear how these differences contribute to the phenotypes observed. What is the causal relationship? Perhaps I am not fully understanding the implications of the experiments (in which case a better explanation would be needed), but it seems that at best, the authors are showing that in GFP sorted phloem cells, there is a larger impact of knocking-out phloem specific function genes compared to non-phloem cells. This would be expected and is only proof of the fact that the GFP-based sort was successful. It seems trivial (again, but perhaps it is not), that one would see a larger effect on a phloem-related gene panel in a mutant with a phloem -related phenotype; compared to other cell types or other genetic backgrounds.

Continuing on this comment, in Fig S9 the authors show that core phloem regulators have reduced chromatin profiles in *smxl4/5* and *smxl5/obe3* mutants in comparison to WT and *smxl5* single mutants. This is nice, but again not surprising as these two double mutants show phloem-related phenotypes, where the WT and *smxl5* single do not. Given these are core regulators, it is to be expected that they would be mis-regulated in the double mutant backgrounds and the authors already showed in Fig1 that their expression is reduced/absent in this background. Although the data is very nice and fits with the previous findings, I am not sure it allows to suggest that these regulators define 'phloem identity' compared to being important regulators at the beginning of the development of this cell type. I do not think this is a semantic remark as the entire selling point of the story is about cell type specification. It would be up to the authors to more clearly indicate why and how their data subscribes to their main conclusion as this

is not the case at the moment in my opinion.

Line 387: 'by creating a distinct chromatin signature important for establishing phloem identity'. This is strong wording. The chromatin signature is not very distinct to begin with and where do the authors show it is important for establishing the identity? There is correlation at best.

Reviewer #3 (Remarks to the Author):

Comments to the Authors

The manuscript submitted by Wallner et al. newly revealed that crucial regulators of SE development SMXLs function together with a putative epigenetic regulator OBE3 to provide the potency differentiating phloem SE. I believe that it offers a very interesting concept to plant vascular community. The authors found an epigenetic regulator OBE3 as one of possible candidates for SMXL4/5 interactors based on Y2H screen. Indeed, genetic analysis with a plenty of combinations including *obe3* mutants beautifully represented the genetic contribution of OBE3 to proper cell division in phloem cell lineage and normal root growth. Moreover, ATAC-seq combined with FACS revealed that SMXL5/OBE3 play important roles in opening chromatin region for phloem related genes. Basically, the results are clearly presented and the conclusion are carefully made based on the results. However, at several points, I feel that there is not enough data to support the conclusion. Here I raised major and minor concerns to improve the manuscript.

Major points

1. *smxl5*; *obe3* double mutants shortened root length and exhibited defects in cell division in phloem cell files. I almost agree that SE development is affected by *smxl5*; *obe3* mutants. However, I feel that the effect of OBE3 on SE differentiation was not fully investigated. Here I recommend the authors to represent the classification of defects in SE differentiation as shown in Fig. 3B. Otherwise, how about the CFDA assay to examine the actual phloem transport in the mutants? Such experiments will be required for demonstrating the contribution of OBE3 to SE differentiation more accurately.
2. Related to the above comment, how is the phenotype of *smxl4*; *smxl5*; *obe3* triple mutant? If OBE3 and SMXLs function in the same genetic pathway, the additional *obe3* mutation may have a little impact on root growth and/or phloem SE development in *smxl4*; *smxl5* mutants.
3. I am wondering why only OBE3 has a specific role in SE development. What do the authors explain about the differences in genetic contribution among OBE3 and other OBEs? Though the authors showed that OBE3 and OBE4 have similar expression patterns in roots, is there any difference in the interaction against SMXL5. I guess, it is possible that the interaction of SMXL5 with other OBEs is easily investigated by FRET-FLIM in *Nicotiana benthamiana*.
4. ATAC-seq can highlight the open chromatin regions which are tightly associated with abundant transcript levels. In other words, it might be just the consequence of epigenetic regulation such as histone modification or DNA methylation. Especially, as the authors wrote that "PHD-finger motifs as carried by the OBE3 protein are known to be epigenetic readers binding to histone H3 tails carrying distinct post-translational modifications such as trimethylation of lysine 4 (H3K4me3) or lysine 9 (H3K9me)", OBE3 potentially controls epigenetic status via histone modification. I am wondering whether phloem genes actually possess repressed histone marks in non-phloem cell lineage. Do the authors have any evidence for that?
5. Phloem SE differentiation is totally suppressed in *smxl4*; *smxl5* mutants. So, I am just

wondering what kinds of cells remain in original phloem cell files in *smxl4;smxl5* mutants? To address this point, the authors can examine the changes in open chromatin regions of other stele markers such as procambium or xylem cell files. In addition, *smxl4;smxl5* mutants had unique open chromatin regions when compared to the WT (Fig. 8). Is it possible that *SMXL5/OBE3* repress the potency to specify the cell fate into other cell types? I think that such an excluding function is important for the establishment of specific cell identity. Therefore, it is worthy analyzing what kinds of genes take more open chromatin region at their TSS sites in *smxl4;smxl5* mutants than in WT.

Minor points

1. OBE3 is broadly expressed in roots. I have a simple question that what happens in the *SMXL5* overexpression lines? Is possible to induce ectopic SE differentiation in roots?

2. Figure 7C, D

Why some individuals of the *obe3*-miRNA mutants have long roots? Is this correlated with the defects in phloem development or with the reduction of mRNA levels of *OBE3*? I understand that *SMXL5*pro-driven *OBE3* can compensate the *smxl5; obe3* double mutant phenotype.

3. Fig. 2E

What does ectopic fluorescent signals in the upper region of the stele mean?

4. Fig. 8

The authors wrote that phloem and non-phloem genes are extracted based on the datasets by Brady et al., (2006). How many genes are used for the comparative analysis? What is the criteria for their selection?

5. Discussion

The authors discussed the role of *SMXL5* as a transcriptional regulator in connection with a transcriptional repressor *TPL*. However, the authors showed the potency that *SMXL5/OBE3* makes the chromatin open in phloem cell lineage. What do the authors think about these opposite behaviors? *SMXL5/OBE3* repress the inhibitory factor of SE differentiation? Please describe the possible roles of *SMXL5/OBE3* in more detail.

6. Line 221

UBI10 should be UBQ10.

7. Line 227

OBE-mGFP should be OBE3-mGFP

Dear Reviewers,

we thank you for the constructive feedback to our first submission which certainly helped to improve the manuscript. As you will see below, resubmission took considerably longer than expected in particular due to technical difficulties when repeating the ATAC-seq analyses.

Overall, we followed most of your suggestions and, in particular, added data on ectopic SMXL5 expression, on the *smxl4;smxl5;obe3* triple mutant and on the nature of the isolated Y2H clones for OBE3. This was in addition to numerous textual changes clarifying several ambiguities and providing more details on experimental conditions.

We hope that we satisfied most of the concerns and are looking forward to another round of constructive feedback.

With our best regards and on behalf of all authors,

Thomas Greb

REVIEWER COMMENTS

Reviewer #1 (Remarks to the Author):

The manual by Wallner et al revealed OBE3 and SMXL5 could interact with each other in nuclei of phloem stem cells and are critical to establish a phloem-specific chromatin profile. They have provided abundant evidences of genetic interaction and protein interaction, and also performed ATAC-seq to illustrate the chromatin accessibility alterations. Overall, this is a comprehensive work to understand the critical roles of OBE3 and SMXLs in the spatial specificity of cell fate decisions. I think the story is likely to be of interest to plant biologists in root developments. However, I have a few concerns, especially on the ATAC-seq parts.

For the ATAC-seq part:

1. Since the authors used SMXL5:H4-GFP as a marker to isolate phloem-associated cells, the SMXL5 promoter therefore are likely more accessible in the GFP⁺ cells than the GFP⁻ cells. However, higher enrichments could be found in GFP⁺ samples of WT, *smxl5* and *smxl5;obe3*, but not in *smxl4;smxl5* (see attached figure), which suggested that either the material is not correct or the isolation of phloem-associated nuclei in *smxl4;smxl5* failed. Many thanks for this comment which allows us to clarify this point. As a response, we rechecked the genotypes of all the lines we used for our ATAC-seq analyses and found them to be correct. We also re-confirmed the phloem defects described for *smxl5;obe3* and *smxl4;smxl5* mutants and rechecked the activity of the SMXL5:H4-GFP transgene in all backgrounds (Fig. S7). This means that there was indeed no mix-up of lines. The fact that the SMXL5 promoter does not show a differential structure in the *smxl4;smxl5* background, can be explained by a preferential effect of both SMXL4 and SMXL5 on its own promoter. For example, SMXL6 and SMXL7 proteins bind to the promoters of the SMXL6 and SMXL7 genes, thereby regulating their activity (Wang et al. 2020, Nature). If we consider SMXL4 and SMXL5 as positive regulators of a phloem-associated chromatin signature, a more condensed SMXL5 conformation is expected. We also would like to mention that the presence of the SMXL5 promoter in our H4-GFP transgene, may result in artefacts for especially this promoter.

2. There're no replicates for ATAC-seq. The differences between samples were limited in arabidopsis. It's better to include replicates to estimate the influence of technical.

As a response to this comment, we worked heavily in the last months on the generation of at least one more dataset comparing wild type, *smx15*, *smx14;smx15*, and *smx15;obe3* lines and this is actually the reason why resubmission took much longer than expected. Unfortunately, although we invested ample resources (human power, funds) into these efforts, we were not able to generate an ATACseq-based comparison of the four genotypes with the same sensitivity as we provided in our initial submission. As the reviewer will see in the included Venn-diagram, many more differential peaks were detected when comparing phloem and non-phloem cells in the initial dataset in comparison to two additional ATACseq

rounds performed in the last months. Although we find several phloem genes in the shared 33 genes (e.g. *CLE25*, *SWEET10*), we believe the new analyses are not conclusive as the number of differential peaks is too low to be meaningful. We can only speculate about the reason for this with the move of our lab into a different building and the associated change of little lab routines as the best explanation we can offer. Obviously, the very same protocol was used in all our attempts, we made sure that the same people performed the experiments (PhD student, lab technician in the sorting facility), and that the same equipment was used (especially the cell sorter). Still, in spite of many attempts, sensitivity (or specificity of the sorting) was substantially lower which unfortunately does not allow us to provide the requested replicates. We would argue, though, that the provided dataset is reliable and more than worth being published. The main reason for this conclusion is that the dataset contains internal controls demonstrating reliability of the data and allowing the estimation of technical noise. The similarity of wild type and *smx15* mutants which do not show phloem defects with regard to their phloem- and non-phloem related chromatin profiles argues for a technically solid and biological relevant analysis. The same is suggested by the similarity of chromatin profiles of the two double mutants which show similar phloem defects. Also the differential profile of phloem-associated genes in wild type and *smx15* mutants and the reduction of these differences in the two double mutants support this conclusion. With GFP-positive and GFP-negative samples for each of the four genotypes, we analysed eight samples by ATACseq in total which, although not being replicates in a strict sense, confirm each other and, as we think, are therefore a valuable source of information for the community. To support our view, we also want to mention that ATAC-seq analyses are often performed without performing replicates and that substantial insight has been obtained by respective studies. Examples are Li et al., 2021, Nature (<https://doi.org/10.1038/s41467-021-27539-3>), Sahu et al., 2022, Nature Gen (<https://doi.org/10.1038/s41467-021-27539-3>) or Krauß et al., 2022, Cancer Res (<https://doi.org/10.1158/0008-5472.CAN-20-3209>).

3. For the ATAC-seq analysis: (1) According to the genome coverage images and the bigwig files in GEO, the qualities of the ATAC-seq samples looked good. But it's better to calculate the FRiP values, since it can also reflect the signal enrichments of each sample. (2) Although the authors have picked up equal reads for the analysis, normalization are still required to deal

with different signal/noise ratio. (3) The authors may quantify the chromatin accessibilities with reads depth in the putative differential peaks and put all samples together to make a heatmap, which would be helpful to classify the differential peaks and also easier for the readers to understand.

(1) We added the FRiP values in Figure 8A. The range of FRiP values in each sample was limited in 0.37 – 0.47, suggesting a similar signal/noise ration among all samples. (2) As suggested, we now updated our analysis pipelines to detect the differential peaks using bam files with reads in only peak regions (merged from the peaks from all samples) to normalize signal/noise ratio among samples. (3) We added a heatmap (Figure 8E) showing the chromatin accessibility of the phloem-specific (SMXL5-GFP+) OCR detected in wild type among all samples analysed. This clearly shows that the phloem-specific OCRs identified in WT are maintained in *smxl5* single mutants, but not in *smxl4;smxl5* and *obe3;smxl5* double mutants. We thank the reviewer for this suggestion.

4. Chromatin accessibilities were highly associated with gene expression. If the authors could perform RNA-seq with the isolated nuclei, it will much better support the conclusions and also reveal the effects of chromatin variations on gene expressions.

Because we were not able to generate another ATACseq dataset with sufficient sensitivity and or/specificity, we abstained from performing an RNAseq experiment which would have required to analyse 24 samples (four genotypes, GFP-positive and GFP-negative samples, three replicates each). With doubts that we are able (at the moment) to reliably generate or analyse phloem-specific nucleus samples, this seemed to be too much of a risk and a waste of resources. This decision was also based on the expectation that, in the best case, we show that phloem-related genes are less active in *smxl4;smxl5* (and *smxl5;obe3*) double mutants which we showed already by analysing respective fluorescent reporters.

Other points:

5. *obe4;smxl5* did not show root deficient phenotypes like *obe3;smxl5*, and OBE3 and OBE4 have similar expression patterns in root (Fig. 6B-6C). How about the protein interaction between OBE4 and SMXL? It may help to explain the functional differentiation between OBE3 and OBE4.

As a response to this comment, we added information on the interaction between SMXL5 and other OBERON proteins in yeast (SupplFig5c, SupplDataset1) and now state “we only detected a genetic interaction between SMXL3/4/5 and OBE3 and not between SMXL4/5 and other OBE family members (Supplementary Figure 5). This was although we isolated, in addition to 24 OBE3 clones, nine clones of OBE2 and three clones of OBE4 in our initial yeast-two hybrid screen (Supplementary Dataset 1) and SMXL5 and OBE2 proteins interacted in yeast cells in independent experiments (Supplementary Figure 5). This indicated a general potential of SMXL5 to interact with OBE proteins, but a functional specificity of OBE3 in phloem formation.” in the text.

6. The localization of OBE3-mCFP in Fig. 4f looks different from those in Fig. 4c and 4m, it would be better to change to a more clear one.

Here, we would like to emphasize that the size, number, and distribution of OBE3 speckles differed a lot from nucleus to nucleus in our experiments, possibly depending on the level of OBE3 protein and its expression dynamics. We believe that it is important to show the whole breadth of signal distribution and, thus, propose to stick to the respective image in Fig. 4f as we do not a good reason to select one type over the other.

7. Fig. 6b, please rotate the root direction to vertical.

This has been changed.

Reviewer #2 (Remarks to the Author):

In the manuscript by Wallner and co-workers, the authors elegantly show how SMXL4/5 act during early phloem development before other known regulators such as OPS, BRX, BAM3 etc. The authors convincingly show how SMXL4/5 as early regulators of phloem development interact with OBE3 in the nucleus. Both proteins overlap in expression domain in the early phloem cells, making it likely that the interaction indeed occurs here. Given the published function of OBERON proteins, the authors go on by undertaking chromatin profiling. Although these results nicely confirm their previous findings, I do feel that the authors are making strong conclusions based on this data (see comments below) and even extrapolate this to claiming that the SMXL4/5-OBE3 complex determines phloem cell fate. I have some reservations about these claims as in some cases, I see a correlation at best. Additionally, the genetic interaction studies are very clear and thoroughly performed, but the conclusions drawn are at moments a bit strong in my opinion as there are equally possible alternative hypothesis based on this data which were not explored. I want to make clear that I am not doubting the data nor the conclusions of the authors throughout the manuscript, but feel that a better argumentation and, at places, a more careful way of describing the conclusions would be needed to ensure the conclusions are in line with the experimental data.

Fig1F and H: what is the GFP signal in the epidermis/cortex cells?

These signals are background signals which, under these conditions, have the tendency to appear at the inner side of the cortex and/or the epidermis. The same signals are visible in Supplementary Figure 1D, F, N showing, together, that the signal is independent from the respective transgene. This is now mentioned in the figure legend of Fig. 1.

Line 126: 'In contrast to...' I do not understand this as the previous sentence is about marker which are also reduced or absent in the smxl double mutant. So, why does this sentence start with 'in contrast' when the observation is the same.

Thanks for this comment, which we agree to. In response, we changed 'in contrast' to 'similar to'.

Line 129: the first reference to Fig S1 starts with panel J. Could these be re-ordered to match the order in the text?

Thanks. The order of panels in Fig S1 has been changed to match their appearance in the main text.

Line 131: the interaction between these components (BRX, OPS, BAM3, CLE45 etc.) is not that trivial to an outsider. Can you integrate a schematic of the pathway somewhere? This would make it easier to follow the genetic experiments that are done and place them in the correct framework. As one example, it is not directly clear why the experiments in the last paragraph related to BAM3 and smxl double mutant are included. Can the authors provide some more background and rationale please? Why were similar experiments not done for the other factors like OPS, BRX etc?

We apologize if our explanation in this regard were not clear enough. To make this part accessible, we expanded the text which now says: "In contrast to the positive phloem regulators OPS, BRX, and CVP2, the phloem-associated BAM3/CLE45 receptor-ligand module counteracts phloem development (Depuydt et al. 2013; Breda et al. 2019). Although the translational BAM3 reporter was less active in smxl4;smxl5 mutants (Figure 1E, F), we

therefore tested genetically the possibility that an hyperactive BAM3/CLE45 signaling pathway is the reason for defective phloem development in *smxl4;smxl5* mutants. Arguing against this possibility, *smxl4;smxl5;bam3* triple mutants showed root growth defects similar to *smxl4;smxl5* double mutants and stimulating the BAM3 pathway by CLE45 treatments had no effect on *smxl4;smxl5* roots (Supplementary Figure 1). Together with the reduced BAM3 reporter activity (Figure 1, E, F), this indicated that, like other phloem-related features, the phloem-associated BAM3/CLE45 pathway was less active in *smxl4;smxl5* mutants and not causing the observed developmental defects.”

Fig1: It would be useful to have insets (enlargement) for the BAM and CVP2 reporters to show if it is expressed or not (similar to what is done for OPS and BRX).

We agree to this comment and have added respective panels to Figure 1.

Line 147: Can the authors explain why the root length in the rescue with the OPS and BAM promoter is significantly longer than WT? Is this over-compensation due to stronger expression compared to the SMXL promoters?

This is indeed an interesting point but difficult to explain. Plants may perform indeed ‘better’ because the OPS and BAM3 promoters may be active in more cells in other organs than the root tip promoting phloem formation and growth overall.

Line 156: what is the rationale for only testing the BRX protein in the complementation of the *smxl* double mutant? Why not OPS, BAM3 etc.?

Thanks for this comment. BAM3 does not makes sense in such an experiment as BAM3 is a negative regulator of phloem development. Because we had difficulties generating an SMXL4:OPS-GFP construct, this line was not generated. Because OPS and BRX act in the same pathway, we concluded that one representative of the two regulators is sufficient for making the point.

Line 166: the genetic data presented in this paragraph is used at the end to conclude that ‘SMXL5 and OPS/BRX genes play roles at different steps during phloem formation with SMXL5 acting upstream’. Can these conclusion be drawn from this genetic data? First, how does it show that SMXL5 acts upstream of OPS/BRX? Next, the fact that the combination of two mutants showing a partial or weak phenotype results in a stronger phenotype can be interpreted in several ways. For example: either they act in the same pathway and knock-out of both makes the overall phenotype worse; or they act in different pathways and the combinatorial effect of perturbing two (unrelated) pathways gives a stronger phenotype compared to the individual ones. Both options are valid in my opinion, making it very difficult to draw conclusions from this type of genetic data. Can the authors explain better how this data unambiguously shows that they work in different steps of phloem formation (based on this genetic data) and how this data shows that SMXL5 acts upstream? Stronger evidence could be that e.g. overexpression of one factor has no effect when the downstream factor is removed. This would show more clearly that one acts upstream of the other.

Thanks for raising this point which allows us to explain and revise our conclusion. Our conclusion is not only based on the genetic data shown in Fig. 3, but also on the fact that OPS and BRX are downregulated in *smxl4;smxl5* mutants but SMXL4 and SMXL5 expression is not altered in *ops* mutants (Fig. 1, Fig. 2). In addition, expressing BRX under the control of SMXL4 promoter in an *smxl4;smxl5* background rescues phloem defects. We agree that data shown in Figure 3 alone would not justify the statements that SMXL functions on the one side and BRX/OPS functions on the other side are distinct but the whole collection of presented data seem to justify this claim. This is what we now make clear at the end of the paragraph. To further address this point, we also generated overexpression lines for SMXL5, as the

reviewer suggested, to see whether SMXL5 is not only required but also sufficient for phloem development. Although we see the functional SMXL5-YFP protein being expressed outside of phloem tissues, no ectopic phloem is formed (Fig. S6) demonstrating that more factors than SMXL5 are required to initiate phloem development. This is in contrast to DOF transcription factors which hold exactly his property (Qian et al., 2022; Nat Plants) and which, therefore, is an important discovery (although the approach therefore cannot be used to answer the question whether SMXL5 is up- or downstream of OPS or BRX).

Fig 4A: what is the + and – control? This is not indicated in the figure or legend. This information is now given in the figure legend.

Fig 4B: for completeness, please show the entire blots in supplemental and a respective loading control.

Entire blots are now shown in Figure S4C. ‘Input’ samples are considered as ‘loading control’ showing the constitution of the protein samples before precipitation. Additional loading controls are not conclusive as protein composition and concentration is very different in the different samples after precipitation.

Fig 4C: what are these nuclear subdomains and why are they perfectly spherical? If associated to e.g. certain genomic regions, I would expect them to be so spherical. Is there any reference in literature to this? Related, why is the circumference of the subdomain from panel C shown as a reduced expression ring in panel D? Is this a bleed-through issue?

The appearance of the speckles is a common phenomenon when transiently expressing interacting proteins. The assumption is that the combination of protein-protein interactions with high levels of protein concentration, often leads to protein agglomerates. Another example is seen in Greb et al., 2007 (<https://doi.org/10.1016/j.cub.2006.11.052>) where this happens with nuclear-localized and chromatin binding proteins completely unrelated to OBE3 and SMXL5. The reason why the circumference of the subdomain from panel C shows a reduced signal ring for mCherry-NLS in panel D may be that OBE3 protein concentration is very high at this place displacing mCherry-NLS.

Line 239: again in this case, the genetic data does not unambiguously show ‘a concerted action of OBE3 and SMXL3, SMXL4 or SMXL5 genes during primary root growth’ as combining partial phenotype mutants from unrelated pathways can also result in stronger phenotypes. Although I am not at all questioning the overall conclusions drawn, the data presented in e.g. Fig5 is not strong evidence for genetic interaction between these players in my opinion as the alternative hypothesis of increased phenotypes upon combination of unrelated pathways remains a viable option.

Thanks for this comment. Although we still would argue that the data show genetic interaction between OBE3 and SMXL genes in the broadest sense, we eliminated the expression ‘genetic interaction’ from this paragraph.

Fig6E-J: what are the massive amount of white dots staining in the outer cell layers being stained? I have seen this is much reduced amount in some mutant lines, but never in this extent and also in the WT roots. Is this a specific modification to the staining?

This is a general phenomenon with our ClearSee method (see also Wallner et al., 2017, <https://doi.org/10.1016/j.cub.2017.03.014> or Depuyt et al., 2013, www.pnas.org/cgi/doi/10.1073/pnas.1222314110). We assume that those granules are plastids which are sometime more or less abundant.

Fig 7B: why is there green staining in the lateral root cap if this is a WT plant? And if this is background fluorescence, then why do we not see this in panel A?

This is, of course, a valid point. The signal in the lateral root cap is background signal which sometime appears due to the mechanical handling of the roots. If the reviewer looks carefully, they will see a similar (although weaker) signal in the lateral root cap in panel A. Importantly, detection of fluorescent SMXL proteins by confocal microscopy is challenging as protein abundance is low and, thus, microscope settings need to be highly sensitive resulting in the sporadic detection of background signals.

Line 375: How can a meta-analysis of chromatin profiles be proof for a comparable function of SMXL4/5 and OBE3 proteins? Related, the authors are using a helicopter-view chromatin profile analysis to draw conclusions about the fact that SMXL and OBE3 determine phloem-related chromatin profiles. I think this is taking the interpretation of the data one step to far. Although I do see small differences in e.g. Fig8B, it is not clear how these differences contribute to the phenotypes observed. What is the causal relationship? Perhaps I am not fully understanding the implications of the experiments (in which case a better explanation would be needed), but it seems that at best, the authors are showing that in GFP sorted phloem cells, there is a larger impact of knocking-out phloem specific function genes compared to non-phloem cells. This would be expected and is only proof of the fact that the GFP-based sort was successful. It seems trivial (again, but perhaps it is not), that one would see a larger effect on a phloem-related gene panel in a mutant with a phloem -related phenotype; compared to other cell types or other genetic backgrounds.

Continuing on this comment, in Fig S9 the authors show that core phloem regulators have reduced chromatin profiles in *smxl4/5* and *smxl5/obe3* mutants in comparison to WT and *smxl5* single mutants. This is nice, but again not surprising as these two double mutants show phloem-related phenotypes, where the WT and *smxl5* single do not. Given these are core regulators, it is to be expected that they would be mis-regulated in the double mutant backgrounds and the authors already showed in Fig1 that their expression is reduced/absent in this background. Although the data is very nice and fits with the previous findings, I am not sure it allows to suggest that these regulators define 'phloem identity' compared to being important regulators at the beginning of the development of this cell type. I do not think this is a semantic remark as the entire selling point of the story is about cell type specification. It would be up to the authors to more clearly indicate why and how their data subscribes to their main conclusion as this is not the case at the moment in my opinion.

We agree with the reviewer that a loss of phloem-related chromatin profile is expected for mutants with reduced phloem identity. However, in our opinion, it is one thing to expect a loss of phloem-related chromatin profiles and to really show experimentally that this is the case. Moreover, the fact that the profile change is the same in *smxl4;smxl5* and in *smxl5;obe3* mutants is, in our view, anything else than trivial. Here, we use chromatin profiling as another phenotyping procedure to characterize the nature of the mutants under investigation and which support functional relationship. If SMXL and OBE3 proteins would act in independent pathways, the chromatin landscape may have been substantially different and our data clearly show that this is not the case. In addition, phloem-specific ATAC-seq has not been done yet and, therefore, the provided ATAC-seq data allow the identification of targets of general phloem regulators. We agree that our wording implied in some cases that there is causality between the function of the investigated proteins and chromatin landscape, which we changed in the revised version of the text. Causality is certainly not possible to infer by the generated data. We still would argue that SMXL and OBE3 promote phloem identity as all phloem-related features tested so far (cell morphology, gene expression, chromatin profile) are absent or much weaker in the respective mutants. In our view, this allows the usage of the term 'cell type specification' to describe the role of the respective regulators.

Line 387: 'by creating a distinct chromatin signature important for establishing phloem identity'. This is strong wording. The chromatin signature is not very distinct to begin with and where do the authors show it is important for establishing the identity? There is correlation at best.

This sentence has been changed in the new version of the manuscript (see also above) and now reads "Based on our findings, we propose that SMXL both proteins fulfil their role in phloem formation by promoting a distinct nuclear signature important for establishing phloem identity."

Reviewer #3 (Remarks to the Author):

Comments to the Authors

The manuscript submitted by Wallner et al. newly revealed that crucial regulators of SE development SMXLs function together with a putative epigenetic regulator OBE3 to provide the potency differentiating phloem SE. I believe that it offers a very interesting concept to plant vascular community. The authors found an epigenetic regulator OBE3 as one of possible candidates for SMXL4/5 interactors based on Y2H screen. Indeed, genetic analysis with a plenty of combinations including obe3 mutants beautifully represented the genetic contribution of OBE3 to proper cell division in phloem cell lineage and normal root growth. Moreover, ATAC-seq combined with FACS revealed that SMXL5/OBE3 play important roles in opening chromatin region for phloem related genes. Basically, the results are clearly presented and the conclusion are carefully made based on the results. However, at several points, I feel that there is not enough data to support the conclusion. Here I raised major and minor concerns to improve the manuscript.

Major points

1. *smxl5; obe3* double mutants shortened root length and exhibited defects in cell division in phloem cell files. I almost agree that SE development is affected by *smxl5; obe3* mutants. However, I feel that the effect of OBE3 on SE differentiation was not fully investigated. Here I recommend the authors to represent the classification of defects in SE differentiation as shown in Fig. 3B. Otherwise, how about the CFDA assay to examine the actual phloem transport in the mutants? Such experiments will be required for demonstrating the contribution of OBE3 to SE differentiation more accurately.

Thanks for this comment. A classification of the defects found in *smxl5;obe3*, *smxl4;obe3* and in *smxl5;obe4* double mutants in comparison to the respective single mutants and to wild type plants has been added to Fig. 6 as panel L. As the reviewer will see, *smxl5;obe3* and *smxl4;obe3* mutants resemble *smxl4;smxl5* mutants with regard to the penetrance of severe phloem defects.

2. Related to the above comment, how is the phenotype of *smxl4; smxl5; obe3* triple mutant? If OBE3 and SMXLs function in the same genetic pathway, the additional *obe3* mutation may have a little impact on root growth and/or phloem SE development in *smxl4; smxl5* mutants. We agree that this is an important experiment. We thus added data comparing root growth of *smxl4;smxl5;obe3* triple mutants with *smxl4;smxl5* and *smxl5;obe3* double mutants (Fig. 5E, F). As the reviewer will see, root length is comparable comparing these genotypes indicating that the three genes regulate the same process at the seedling stage. Of note, at later stages

triple mutants are lethal in contrast to double mutants (indicated in the figure legend). Because the reason for this is difficult to determine, we think that addressing this point goes beyond the scope of this study.

3. I am wondering why only OBE3 has a specific role in SE development. What do the authors explain about the differences in genetic contribution among OBE3 and other OBEs? Though the authors showed that OBE3 and OBE4 have similar expression patterns in roots, is there any difference in the interaction against SMXL5. I guess, it is possible that the interaction of SMXL5 with other OBEs is easily investigated by FRET-FLIM in *Nicotiana benthamiana*.

This is an interesting question which, however, we cannot answer based on the data we collected within this study. All OBE genes seem to be expressed ubiquitously and we isolated clones of OBE2 and OBE4 during our Y2H screen (now shown in SupplDataset1) and see interaction between SMXL5 and OBE2 in independent Y2H experiments (now shown in SupplFig5), indicating a general interaction between SMXL proteins and OBE proteins in transient and heterologous expression systems. Because we do not see a genetic contribution of OBE1, OBE2 and OBE4, we think that further experiments are therefore not conclusive.

4. ATAC-seq can highlight the open chromatin regions which are tightly associated with abundant transcript levels. In other words, it might be just the consequence of epigenetic regulation such as histone modification or DNA methylation. Especially, as the authors wrote that “PHD-finger motifs as carried by the OBE3 protein are known to be epigenetic readers binding to histone H3 tails carrying distinct post-translational modifications such as trimethylation of lysine 4 (H3K4me3) or lysine 9 (H3K9me)”, OBE3 potentially controls epigenetic status via histone modification. I am wondering whether phloem genes actually possess repressed histone marks in non-phloem cell lineage. Do the authors have any evidence for that?

This again is an interesting point which we are currently working on in the context of a PhD project. We do not, however, have any robust data in this regard already and think that this is beyond the scope of this study.

5. Phloem SE differentiation is totally suppressed in *smxl4;smxl5* mutants. So, I am just wondering what kinds of cells remain in original phloem cell files in *smxl4;smxl5* mutants? To address this point, the authors can examine the changes in open chromatin regions of other stele markers such as procambium or xylem cell files. In addition, *smxl4;smxl5* mutants had unique open chromatin regions when compared to the WT (Fig. 8). Is it possible that SMXL5/OBE3 repress the potency to specify the cell fate into other cell types? I think that such an excluding function is important for the establishment of specific cell identity. Therefore, it is worthy analyzing what kinds of genes take more open chromatin region at their TSS sites in *smxl4;smxl5* mutants than in WT.

Thanks for this comment. To address this point, we did the same analyses we did for phloem-related and non-phloem-related genes also for xylem-related genes. As a result, no differential chromatin conformation could be detected for xylem-related genes comparing GFP-positive and GFP-negative samples or comparing the different genetic backgrounds. This indicates that there was no particular alteration of chromatin signatures around genes associated with other vascular tissues in *smxl4;smxl5* and *smxl5;obe3* double mutants arguing for a specific role of SMXL5 and OBE3 in phloem cells and in targeting phloem-related genes. The new data are now shown in Fig. S11 and in Suppl dataset 6.

Minor points

1. OBE3 is broadly expressed in roots. I have a simple question that what happens in the SMXL5 overexpression lines? Is possible to induce ectopic SE differentiation in roots? See above. Effects of ectopic SMXL5 expression are now shown in Fig. S6.

2. Figure 7C, D

Why some individuals of the obe3-miRNA mutants have long roots? Is this correlated with the defects in phloem development or with the reduction of mRNA levels of OBE3? I understand that SMXL5pro-driven OBE3 can compensate the smx15; obe3 double mutant phenotype.

This is a common observation when miRNA or RNAi approaches are performed. The effect on target genes can be quite variable in different plants or lines (Schwab et al., 2006, <https://doi.org/10.1105/tpc.105.039834>; Kerschen et al., 2004, <https://doi.org/10.1016/j.febslet.2004.04.043>).

3. Fig. 2E

What does ectopic fluorescent signals in the upper region of the stele mean?

See above. Such a signal sometimes appears due to the mechanical handling of the roots and the associated tissue damage. Importantly, detection of fluorescent SMXL proteins by confocal microscopy is challenging as protein abundance is low and, thus, microscope settings need to be highly sensitive resulting in the sporadic detection of background signals. The background signal is now accordingly described in the figure legend.

4. Fig. 8

The authors wrote that phloem and non-phloem genes are extracted based on the datasets by Brady et al., (2006). How many genes are used for the comparative analysis? What is the criteria for their selection?

The selected genes are depicted in SupplDataset 6. The selection was performed in Brady et al. based on their differential expression in tissue-specific transcriptome analysis.

5. Discussion

The authors discussed the role of SMXL5 as a transcriptional regulator in connection with a transcriptional repressor TPL. However, the authors showed the potency that SMXL5/OBE3 makes the chromatin open in phloem cell lineage. What do the authors think about these opposite behaviors? SMXL5/OBE3 repress the inhibitory factor of SE differentiation? Please describe the possible roles of SMXL5/OBE3 in more detail.

Thanks, this aspect was indeed missing. Such a discussion is now included at the end of the discussion section.

6. Line 221

UBI10 should be UBQ10.

This was corrected.

7. Line 227

OBE-mGFP should be OBE3-mGFP

Thanks, this was corrected.

Reviewer #1 (Remarks to the Author):

Thanks for the authors' responses. The authors have made great efforts to answer my concerns. However, I am sorry that I am not convinced by the response of the first two points based on the following reasons: (1) According to the method parts, reads were aligned to the original TAIR10 genome, therefore the reads from SMXL5:H4-GFP would also be aligned to SMXL5 promoter. The more copy number could explain the higher read depth than the neighboring regions in all GFP+ and GFP- transgenic tracks except for smxl4;smxl5 (see the attached figure in the first round review). This suggests that the SMXL5:H4-GFP is likely missed in smxl4;smxl5; (2) From Fig. S7, we could see the GFP signals in smxl4;smxl5 were similar with that in smxl5;obe3, indicating similar promoter activities in the two lines. This would deny the authors' speculations that SMXL5 were not activated due to the absence of SMXL4 and SMXL5. Considering the authors could not replicate the ATAC-seq, I would have to doubt the quality and results of these parts. I think the author may need to add at least one more replicate for these specific pairs to confirm the results.

Reviewer #2 (Remarks to the Author):

In this revised manuscript, the authors have taken a considerable effort to address the reviewers comments. Related to my previous remarks, the authors have responded to and addressed all my concerns to the best of their abilities. Although I leave it up to the authors to decide how to define this in the final version of the manuscript, I would like to make a final suggestion regarding the comment on how chromatin profiles could provide evidence on stating that factors have a role in determining phloem identity.

Although I appreciate the explanation of the authors describing how they see that the data contributes to making this statement, one can wonder what defines tissue identity in the first place. I am sure many ideas exist, but for me this is probably the combination of the different transcriptional responses a cell has at one moment in time due to the plethora of external stimuli it receives. This is to a large amount determined by its position in the tissue. If this definition makes some sense to the authors, then any mutant affecting phloem development will obviously change the chromatin/expression profile. In this train of thoughts, any mutant affecting the phloem transcriptome (which in my view is likely any mutant affecting phloem development) would thus be controlling phloem identity in the authors opinion. With the data present, we cannot address this point, as one would then need to profile unrelated phloem regulators to see if these behave different or not. Clearly this would be outside the scope of this manuscript, but I hope the authors see where I am trying to go with this remark.

I do feel this is not simply a semantic discussion, but goes to the core of the paper and the message it brings. Again, I do not disagree with the authors and the data is strong and supportive, but I would myself be more careful in interpreting the combination of results as to stating these factors control phloem identity. This could be something to suggest in the discussion, but having it even in the title is a bold move in my opinion.

Another example catching my attention is the legend to Fig1 where it states that genes are 'less active' in the mutant background, while what is shown is a reduced protein abundance. Is this really the same?

As mentioned earlier. These are on-going discussions in the field and I am sure the authors will agree with me that we actually do not know for sure what defines a tissue identity. As such, I do not want to make strong statements from my side neither. One thing the authors might want to consider is whether they want to make such strong claims (including the title) for future reference. This manuscript contains very nice data and I would support publication also without these (strong) claims, so it is worth considering from the authors side to slightly tone down the statements.

Reviewer #3 (Remarks to the Author):

Thank you very much for adding the new genetic data and for responding to my comments. The

authors almost addressed my concerns. Moreover, the authors toned down the conclusion regarding chromatin regulation by SMXLs and OBEs. I have nothing more to say at this stage.

Dear Reviewers,

we thank again for your comments and critical thoughts. In response, we slightly changed the wording in the title, abstract, and main text by replacing 'phloem identity' by 'phloem development'. With regard to the concern toward the ATAC-seq data on the SMXL5 promoter, we believe, as we outline below, that we have valid reasons to conclude that the presented data are technically sound and support the main statements made in this study.

With our best regards and on behalf of all the authors,

Thomas Greb

Reviewer #1 (Remarks to the Author):

Thanks for the authors' responses. The authors have made great efforts to answer my concerns. However, I am sorry that I am not convinced by the response of the first two points based on the following reasons: **(1)** According to the method parts, reads were aligned to the original TAIR10 genome, therefore the reads from SMXL5:H4-GFP would also be aligned to SMXL5 promoter. The more copy number could explain the higher read depth than the neighboring regions in all GFP+ and GFP- transgenic tracks except for *smxl4;smxl5* (see the attached figure in the first round review). This suggests that the SMXL5:H4-GFP is likely missed in *smxl4;smxl5*; **(2)** From Fig. S7, we could see the GFP signals in *smxl4;smxl5* were similar with that in *smxl5;obe3*, indicating similar promoter activities in the two lines. This would deny the authors' speculations that SMXL5 were not activated due to the absence of SMXL4 and SMXL5. Considering the authors could not replicate the ATAC-seq, I would have to doubt the quality and results of these parts. I think the author may need to add at least one more replicate for these specific pairs to confirm the results.

We again thank for these comments which allow us to clarify this point. We firmly believe that the SMXL5:H4-GFP transgene was present in the *smxl4;smxl5* background and that enrichment of phloem nuclei and subsequent ATAC-seq was successful for the following reasons. **a)** We thoroughly tested genotypes including the presence of the transgenes during our experiments by PCR on genomic DNA and through the antibiotics resistance marker. There was never a case in the final lines used for the ATAC-seq experiments in which the transgene was missing. **b)** Analyses using the confocal microscope always detected a fluorescence signal as now shown in Fig. S7. **c)** The plots generated during nuclear sorting for the performed ATAC-seq experiments (Fig. S7), demonstrated the presence of GFP-positive nuclei, similar to the other lines carrying the transgene and unlike a wild type control not carrying such a transgene (see P4 gate in the plots). **d)** Except for the SMXL5 promoter, genomic regions of other phloem genes still show a mild difference in compactness comparing GFP+ and GFP- samples in *smxl4;smxl5* as shown in Supplemental Figure 9 and 10. **e)** The similarity between the chromatin profiles obtained from *smxl4;smxl5* and *smxl5;obe3* mutants, and their difference to chromatin profiles from wild type and *smxl5* single mutants, furthermore indicates the validity of the *smxl4;smxl5* data set. Altogether, this demonstrates that GFP-based enrichment of phloem nuclei from *smxl4;smxl5* plants and their analysis by ATAC-seq was successful which, as we believe, is the central point of this experiment.

We only can speculate why the SMXL5 promoter region does not show a different read alignment depth comparing GFP+ and GFP-samples from *smxl4;smxl5* plants and why read alignment to the

SMXL5 promoter is not enhanced overall in comparison to other samples. As we mentioned before, the absence of both SMXL5 and SMXL4 in this background may have a particular effect on the compactness of the SMXL5 promoter and the same effect on transcription of the transgene like the absence of SMXL5 and OBE3. As SMXL proteins directly influence the expression of their own genes in other cases (Wang et al., 2020, Nature), the SMXL promoter might be particularly sensitive in this regard. Moreover, it is important to mention that we transformed transgenes independently to each line, and this might cause different effects to the inserted transgenes. Although the expression of H4-GFP was checked by fluorescence analysis and showed similar levels and patterns as *smxl5;obe3* mutants, the chromatin state of SMXL5:H4-GFP transgenes might be different due to their distinct genomic positions.

Reviewer #2 (Remarks to the Author):

In this revised manuscript, the authors have taken a considerable effort to address the reviewers comments. Related to my previous remarks, the authors have responded to and addressed all my concerns to the best of their abilities. Although I leave it up to the authors to decide how to define this in the final version of the manuscript, I would like to make a final suggestion regarding the comment on how chromatin profiles could provide evidence on stating that factors have a role in determining phloem identity.

Although I appreciate the explanation of the authors describing how they see that the data contributes to making this statement, one can wonder what defines tissue identity in the first place. I am sure many ideas exist, but for me this is probably the combination of the different transcriptional responses a cell has at one moment in time due to the plethora of external stimuli it receives. This is to a large amount determined by its position in the tissue. If this definition makes some sense to the authors, then any mutant affecting phloem development will obviously change the chromatin/expression profile. In this train of thoughts, any mutant affecting the phloem transcriptome (which in my view is likely any mutant affecting phloem development) would thus be controlling phloem identity in the authors opinion. With the data present, we cannot address this point, as one would then need to profile unrelated phloem regulators to see if these behave different or not. Clearly this would be outside the scope of this manuscript, but I hope the authors see where I am trying to go with this remark.

I do feel this is not simply a semantic discussion, but goes to the core of the paper and the message it brings. Again, I do not disagree with the authors and the data is strong and supportive, but I would myself be more careful in interpreting the combination of results as to stating these factors control phloem identity. This could be something to suggest in the discussion, but having it even in the title is a bold move in my opinion.

Another example catching my attention is the legend to Fig1 where it states that genes are 'less active' in the mutant background, while what is shown is a reduced protein abundance. Is this really the same?

As mentioned earlier. These are on-going discussions in the field and I am sure the authors will agree with me that we actually do not know for sure what defines a tissue identity. As such, I do not want to make strong statements from my side neither. One thing the authors might want to consider is whether they want to make such strong claims (including the title) for future reference. This manuscript contains very nice data and I would support publication also without these (strong) claims, so it is worth considering from the authors side to slightly tone down the statements.

We appreciate these thoughtful comments and agree with the reviewer that the term ,identity' is ambiguous and possibly hard to justify when using it in its common sense. Indeed, this term can only be applied if respective mutants are compared to other phloem mutants in which only a subset of phloem features is absent or with mutants which do not develop phloem, but show similar changes in chromatin structures. We used the term ,identity', because of the comprehensive effect of SMXL4/5-deficiency in this regard and as the role of chromatin regulators is traditionally associated with this term. With the recent discovery that higher order mutants of DOF transcription factors also show a fundamental block in phloem development (Qian et al., 2022, Nat Plants) and act possibly upstream of SMXL genes (Miyashima et al., 2019, Nature), there would now be interesting genetic material available for such a comparison. In response to this comment, we changed our wording in the title, the abstract and throughout the text from ,phloem identity' to ,phloem development'. We believe that this change indeed makes the wording more solid.

We also changed the legend of Fig. 1 along the lines suggested by the reviewer. Instead of "Phloem-related genes are less active in *smxl4;smxl5* mutants." It now reads "Phloem-related transcriptional and translational reporters are less active in *smxl4;smxl5* mutants."

Reviewer #3 (Remarks to the Author):

Thank you very much for adding the new genetic data and for responding to my comments. The authors almost addressed my concerns. Moreover, the authors toned down the conclusion regarding chromatin regulation by SMXLs and OBEs. I have nothing more to say at this stage.

We thank the reviewer for the effort and support.

Reviewer #1 (Remarks to the Author):

I appreciate the authors' efforts in explaining the exist of H4-GFP signals. A minor suggestion, a genotyping result to show the exist of SMXL5:H4-GFP, especially the SMXL5 promoter would be useful to answer the concern. Otherwise, this suggestion is not necessary for the manuscript be considering for publication.

Reviewer #2 (Remarks to the Author):

Before delivering this additional review, I would like to stress that I was specifically tasked by the senior editor to evaluate the discussion regarding the points raised by reviewer1.

As a first general remark, I am sure we can all agree that any published data should be reproducible. This is indeed a basic requirement in science. This being said, I am also in touch with the reality of doing experiments where there is an inherent variability which is not always easy to explain.

It is clear that reviewer1 has a very good point questioning the reproducibility of the experiments. The authors on the other hand have invested a large amount of time to do exactly so but the overall quality of these repeats was lower compared to the initial experiments. In order to prevent what could quickly become a very semantic discussion, I would like to propose the following. The authors could include all this information about the repeats and the lower quality in the supplementary materials and methods and simply explain what was done and give the arguments as to why they believe this is still a valid dataset and conclusions. This way, any reader will see what was done and can interpret these results in the correct framing.

In my humble opinion, I do believe these results are very interesting to the community and the authors make a strong and realistic case as to why they believe their data is trustworthy. As such, to me, the best way forward is to present all available information in an open and transparant way.

I hope this intermediate solution can be acceptable for both sides.

Reviewer #3 (Remarks to the Author):

I agree with the comment of Reviewer2. I have the impression that cell identity is not yet strictly defined in the context of development. We need to discover factors like pioneer factors that have the potential to directly change the epigenetic state and can trigger cell differentiation. I believe that this manuscript is an important paper that provides a stepping stone to this goal. I don't see any problem with it, since the authors toned down from "phloem identity" to "phloem development". I also support the publication of this manuscript.

Reviewer #1 (Remarks to the Author):

I appreciate the authors' efforts in explaining the exist of H4-GFP signals. A minor suggestion, a genotyping result to show the exist of SMXL5:H4-GFP, especially the SMXL5 promoter would be useful to answer the concern. Otherwise, this suggestion is not necessary for the manuscript be considering for publication.

As a response, we detected the *SMXL5:H4-GFP* transgene in the respective lines by performing a PCR on genomic DNA. The result is shown below. The transgene is indeed detected in all individual plants which were analysed (five for each line).

Product length: 579 bp PRIMERS: CEB1for13 (SMXL5pro) + GFPprev 1% agarose gel

Reviewer #2 (Remarks to the Author):

Before delivering this additional review, I would like to stress that I was specifically tasked by the senior editor to evaluate the discussion regarding the points raised by reviewer1.

As a first general remark, I am sure we can all agree that any published data should be reproducible. This is indeed a basic requirement in science. This being said, I am also in touch with the reality of doing experiments where there is an inherent variability which is not always easy to explain.

It is clear that reviewer1 has a very good point questioning the reproducibility of the experiments. The authors on the other hand have invested a large amount of time to do exactly so but the overall quality of these repeats was lower compared to the initial experiments. In order to prevent what could quickly become a very semantic discussion, I would like to propose the following. The authors could include all this information about the repeats and the lower quality in the supplementary materials and methods and simply explain what was done and give the arguments as to why they believe this is still a valid dataset and conclusions. This way, any reader will see what was done and can interpret these results in the correct framing.

In my humble opinion, I do believe these results are very interesting to the community and the authors make a strong and realistic case as to why they believe their data is trustworthy. As such, to me, the best way forward is to present all available information in an open and transparent way.

I hope this intermediate solution can be acceptable for both sides.

Thanks for these suggestions. As a response considerations about the absence of replicates in a strict sense have now been included into the Methods section.

Reviewer #3 (Remarks to the Author):

I agree with the comment of Reviewer2. I have the impression that cell identity is not yet strictly defined in the context of development. We need to discover factors like pioneer factors that have the potential to directly change the epigenetic state and can trigger cell differentiation. I believe that this

manuscript is an important paper that provides a stepping stone to this goal. I don't see any problem with it, since the authors toned down from "phloem identity" to "phloem development". I also support the publication of this manuscript.

Thanks for these supporting remarks.